# Zinc ion flux during mammalian sperm capacitation

Karl Kerns[1], Michal Zigo[1,2], Erma Z. Drobnis[3], Miriam Sutovsky[1] & Peter Sutovsky[1,3]

Sperm capacitation, the ultimate maturation event preparing mammalian spermatozoa for fertilization, was first described in 1951, yet its regulatory mechanisms remain poorly understood. The capacitation process encompasses an influx of bicarbonate and calcium ions, removal of decapacitating factors, changes of pH and sperm proteasomal activities, and the increased protein tyrosine phosphorylation. Here, we document a novel biological phenomenon of a unique zinc ($Zn^{2+}$) ion redistribution associated with mammalian sperm in vitro capacitation (IVC). Using image-based flow cytometry (IBFC), we identified four distinct types of sperm zinc ion distribution patterns (further zinc signature) and their changes during IVC. The zinc signature was altered after sperm capacitation, reduced by proteasomal inhibitors, removed by zinc chelators, and maintained with addition of external $ZnCl_2$. These findings represent a fundamental shift in the understanding of mammalian fertilization, paving the way for improved semen analysis, in vitro fertilization (IVF), and artificial insemination (AI).

[1] Division of Animal Sciences, University of Missouri, Columbia, MO 65211-5300, USA. [2] Laboratory of Reproductive Biology, Institute of Biotechnology, Czech Academy of Sciences, v.v.i., 25242 Vestec, Czech Republic. [3] Department of Obstetrics, Gynecology and Women's Health, University of Missouri, Columbia, MO 65211-5300, USA. Correspondence and requests for materials should be addressed to P.S. (email: SutovskyP@missouri.edu)

A zinc spark (an exocytotic event releasing billions of zinc ions) issued from the oocyte induced by the spermatozoa at fertilization[1] is implicated as a novel biomarker of mammalian embryo quality and developmental potential[2]. To date, zinc ion ($Zn^{2+}$) fluxes have not been well characterized in mammalian spermatozoa, though sperm-flagellar voltage-gated proton channel HVCN1, negatively regulated by $Zn^{2+}$, has been implicated as the main proton extrusion mechanism during mammalian sperm capacitation[3]. This channel regulates intracellular pH and consequently is thought to be responsible for $Ca^{2+}$ entry via opening of the CatSper channel, all of these events coinciding with PKA activation and the hallmark increase of protein tyrosine phosphorylation during boar sperm capacitation[4,5]. Further, the 26S proteasome, a multi-subunit ubiquitin-dependent protease, regulates fertilization at multiple steps from spermiogenesis to sperm penetration of the oocyte zona pellucida (ZP)[6] including certain aspects of sperm capacitation[7]. In particular, the A-kinase anchoring protein AKAP3 is degraded by the ubiquitin-proteasome system (UPS) during bull sperm capacitation[8], and the E1-type ubiquitin-activating enzyme (UBA1) inhibitor PYR-41 alters acrosomal remodeling. Additionally, proteasomal inhibitors hinder the capacitation-associated shedding of acrosin-inhibitor serine peptidase inhibitor kazal type 2 (SPINK2) and spermadhesin AQN1[9] in the boar (see review[7]).

Here, we use image-based flow cytometry (IBFC) to document four distinct types of sperm zinc signature and their changes during in vitro capacitation (IVC) of domestic boar, bull, and human spermatozoa, altered after sperm capacitation, further reduced under proteasome inhibiting IVC conditions, removed by zinc chelators, and maintained with addition of external $ZnCl_2$. The zinc shield established by the oocyte following fertilization could derail such sperm zinc signaling as an added barrier to pathological polyspermic fertilization. This all together supports a new role of zinc ions during capacitation and fertilization. Such findings represent a fundamental shift in the understanding of mammalian fertilization, paving the way for a more accurate semen analysis to ameliorate the methodology of in vitro fertilization (IVF) and artificial insemination (AI).

## Results

**Mammalian spermatozoa possess four distinct zinc signatures.** We used state-of-the-art IBFC and epifluorescence microscopy to trace the sperm zinc signature using Zn-probe FluoZin™-3 AM (FZ3), DNA stain Hoechst 33342, acrosomal remodeling detecting lectin PNA (*Arachis hypogea*/peanut agglutinin) conjugated to Alexa Fluor™ 647 (PNA-AF647), and live/dead cell, plasma membrane integrity reflecting DNA stain propidium iodide (PI), which is taken up exclusively by cells with a compromised/remodeled plasma membrane. The IBFC, which combines the fluorometric capabilities of conventional flow cytometry with high speed-multi-channel image acquisition, proved to be advantageous due to the high presence of $Zn^{2+}$ in sperm cytoplasmic droplets and seminal debris, which otherwise would distort traditional flow cytometry results. We developed a unique gating and masking strategy to ensure unbiased data analysis (Supplementary Fig. 1). Analyses were performed using the initial, pre-sperm-rich fraction of ejaculates, which had highest sperm viability/plasma membrane integrity, repeatability, and sensitivity to proteasomal inhibition compared to secondary, sperm-rich fraction that appeared more prone to spontaneous capacitation and loss of plasma membrane integrity.

We identified four distinct types of sperm zinc signatures conserved across boar (Fig. 1a–d), bull (Fig. 1e–h), and human spermatozoa (Fig. 1i–l): high $Zn^{2+}$ presence in the sperm head and whole-sperm tail (signature 1; Fig. 1a, e, i), medium-level (based on relative intensity of fluorescence in FlowSight measurements) $Zn^{2+}$ presence in both the sperm head and

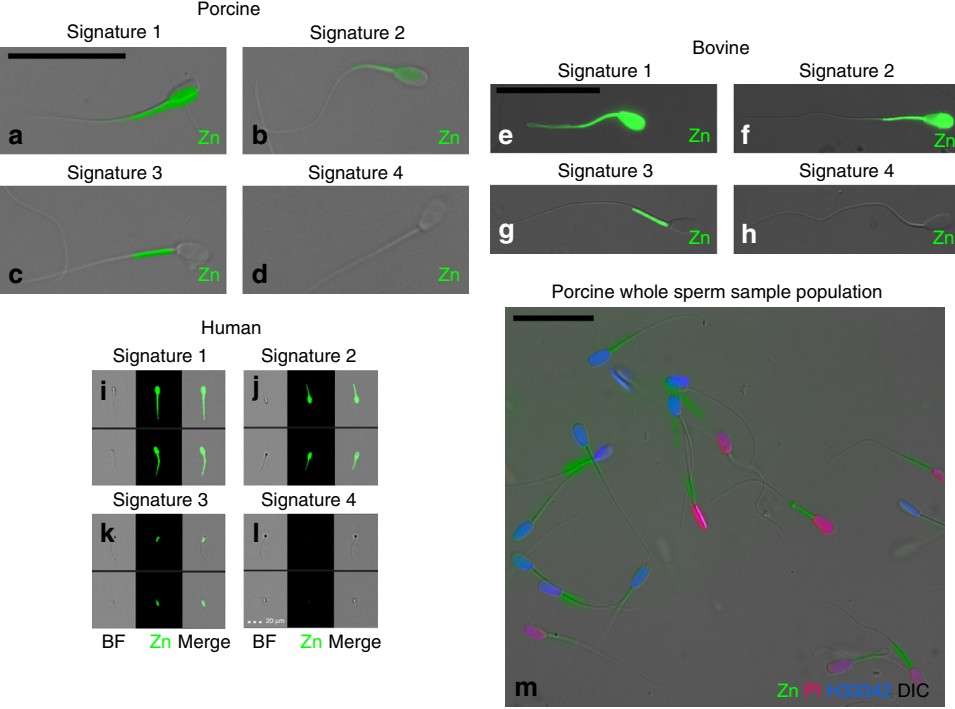

**Fig. 1** Mammalian sperm zinc signature. **a–d** Epifluorescence microscopy of boar sperm zinc signature (green). **e–h** Epifluorescence microscopy of bull sperm zinc signature. **i–l** FlowSight image gallery of human sperm zinc signature (scale bar: 20 μm). **m** Boar spermatozoa after 72 h of storage in Beltsville thaw solution (BTS semen extender) show varied zinc signatures. Imprecise fluorescent to bright-field overlay illustrates motile status (all scale bars: 25 μm, unless noted)

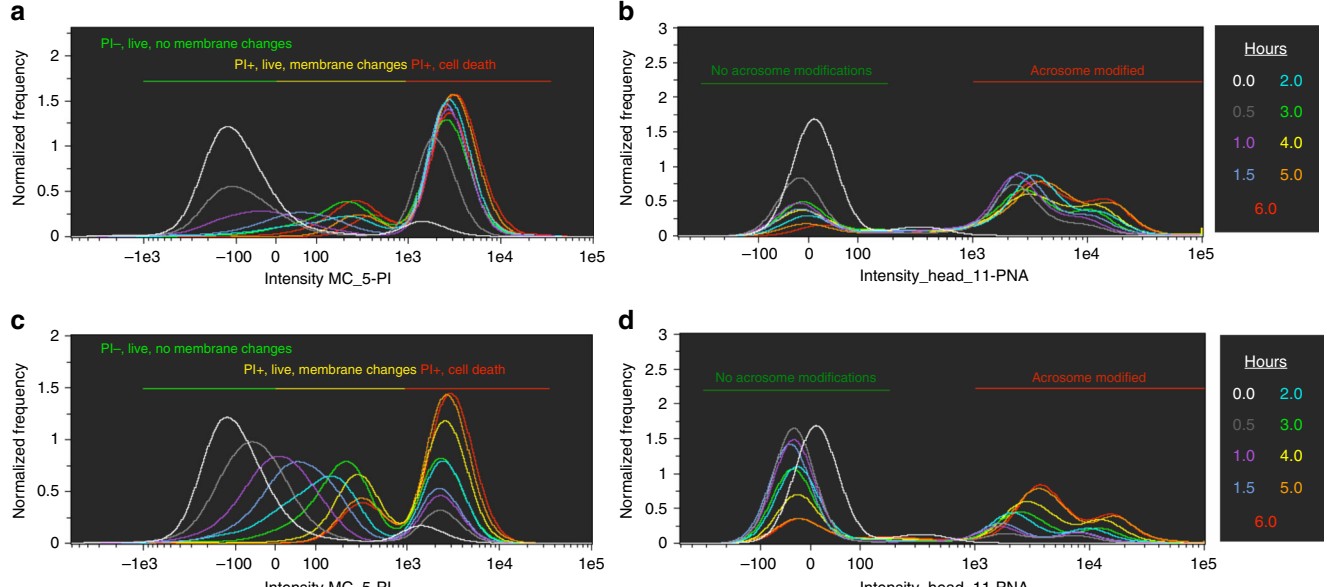

**Fig. 2** Zinc signature time course with high vs. low bicarbonate IVC. Time-lapse recordings of zinc signature during IVC in high, 15 mM sodium bicarbonate media: (**a**) plasma membrane changes as identified by PI status, distinguishing between IVC-induced PI$^+$ subpopulations (PI$^+$ live with plasma membrane changes vs. PI$^+$ cell death). **b** Acrosomal modifications. Time course of zinc signature modification in low, 2 mM sodium bicarbonate IVC media: (**c**) plasma membrane changes, and **d** acrosomal modifications (corresponding histogram color code for time points in figure legend)

sperm tail midpiece (signature 2, spermatozoa undergoing capacitation; Fig. 1b, f, j), Zn$^{2+}$ presence in the midpiece only (signature 3/capacitated state signature in spermatozoa that underwent capacitation and may be dying; Fig. 1c, g, k), and no Zn$^{2+}$ presence (signature 4, spermatozoa with compromised/ remodeled plasma membrane; Fig. 1d, h, l). Spermatozoa after 72 h of storage in Beltsville thaw solution (BTS semen extender) show varied zinc signatures (Fig. 1m).

**Zinc signature is indicative of capacitation status in vitro.** A drawback to commonly used 15 mM sodium bicarbonate IVC media is rapid sperm death (as compared to in vivo sequential capacitation[10]), illustrated in the time course study by a shift to PI$^+$ cell death flow cytometry gating (Fig. 2a) and rapid acrosomal modification (Fig. 2b). In the interest of emulating in vivo sperm lifespan and sequential capacitation as a fertility diagnostic method, we used a previously described capacitation medium[11] with low (2 mM) sodium bicarbonate and increased sodium pyruvate (5 mM) that prolonged sperm viability (Fig. 2c) and elicited similar hyperactivation (Supplementary Movie 1) while achieving hallmark acrosomal modification (Fig. 2d; discussed further in Methods, in vitro capacitation section).

Most spermatozoa in zinc signature 1 and 2 states had no capacitation-like acrosomal remodeling (93.0 ± 6.8% and 95.0 ± 2.6%, data presented as mean ± s.d.; 10,000 cells analyzed per treatment, $n = 3$ biological replicates) compared to zinc signature 3 and 4 (11.1 ± 5.8% and 7.0 ± 9.9%; $P < 0.0001$, as determined by the general linear model (GLM) procedure). Capacitation-like acrosomal remodeling was most prevalent with zinc signatures 3 and 4 (81.0 ± 8.5% and 62.2 ± 12.9%) compared to zinc signatures 1 and 2 (4.0 ± 4.7% and 3.4 ± 2.9%; $P < 0.0001$, as determined by the GLM procedure). Acrosome exocytosis occurred within the subpopulation of spermatozoa with zinc signature 4 (30.7 ± 3.0%) and was greater than zinc signatures 1, 2, and 3 (3.0 ± 2.6%, 1.6 ± 1.2%, 7.9 ± 2.9%; $P < 0.001$, as determined by the GLM procedure; Fig. 3a; Table 1). As sperm plasma membrane integrity decreased, signaled by increased PI labeling, the zinc patterns progressed to

signatures 3 and 4 (Fig. 3b). Hyperactivated spermatozoa, capable of recognizing and binding the oocyte ZP have zinc signature 2 (Supplementary Movies 2–4), in which the transition from signature 1 to 2 occurs within the first 30–60 min of IVC (Supplementary Fig. 2m).

**26S proteasome modulates zinc signature capacitation shift.** Fresh, ejaculated boar spermatozoa mostly had signature 1, (83.8 ± 3.1%; data presented as mean ± s.e.m.; 10,000 cells analyzed per treatment, $n = 3$ biological replicates; Fig. 4a, e; Table 2). A small portion of spermatozoa incubated in non-IVC media for 4 h at 37 ° C progressed to signature 2 (Fig. 4b) as compared to spermatozoa in the same media incubated at room temperature to emulate the conditions of AI (Fig. 4a), suggesting that some spermatozoa undergo temperature-induced, early-stage capacitation. When proteasome inhibitor MG-132 was added to IVC conditions to reduce sperm proteasome activity as previously described[12,13], a significantly higher portion of spermatozoa retained signature 1 when using the pre-sperm-rich fraction (Fig. 4c, d, e) as compared to IVC+ vehicle ($P = 0.0271$; when signatures 1 and 2 combined $P = 0.0008$, as determined by Duncan's multiple range test). After 4 h of IVC, the zinc signature changed to mostly signature 3 (49.4 ± 7.9%), with a small portion of spermatozoa having signature 2 (31.3 ± 12.3%; Fig. 4c). Remarkably, manipulation of sperm Zn$^{2+}$ content during IVC reset the zinc signature (Fig. 4f, h, i). Spermatozoa retained signature 1 with addition of 1 mM ZnCl$_2$ to IVC medium (Fig. 4f). Cell-permeant Zn$^{2+}$ chelator N,N,N′,N′-Tetrakis(2-pyridinylmethyl)-1,2-ethanediamine (TPEN) removed a majority of FZ3 fluorescence to signature 3 and 4 states (Fig. 4g) compared to TPEN vehicle (Supplementary Fig. 3h), but TPEN did not reduce Fluo-4 calcium probe intensity compared to vehicle (Supplementary Fig. 3i). Remaining FZ3 fluorescence is likely due to Zn presence, but ions tightly bound within the mitochondrial sheath, as zinc has been previously detected there by electron microscopy[14]. With the exception of the midpiece, zinc ions appeared to be associated with the sperm surface, as the stepwise extraction removed Zn$^{2+}$ tracer fluorescence early in treatment procedure (Supplementary Fig. 3j–l).

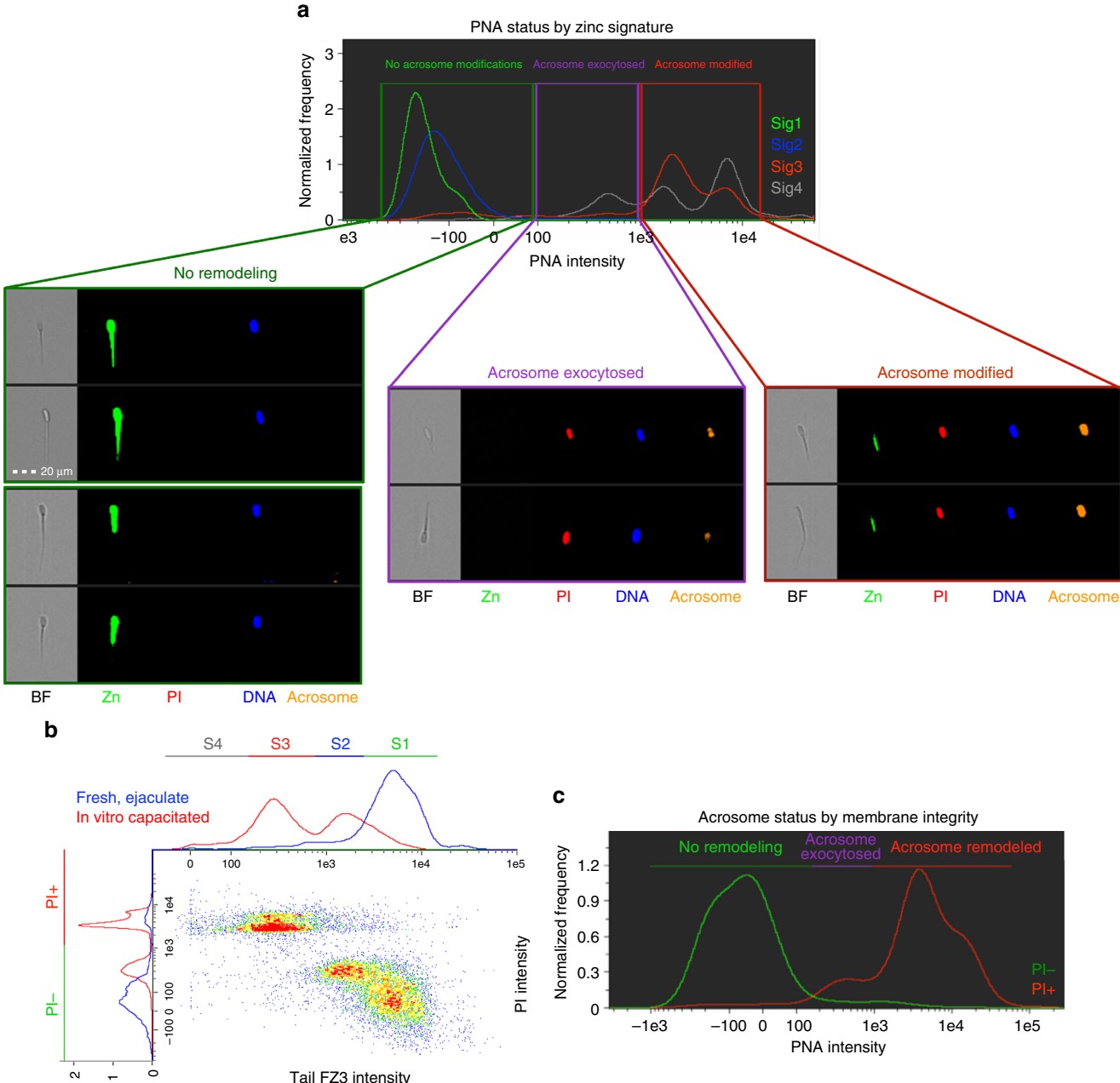

**Fig. 3** Acrosomal status and membrane integrity of zinc signature. **a** Most spermatozoa in zinc signature 1 and 2 states had no capacitation-like acrosomal remodeling compared to zinc signature 3 and 4 ($P < 0.0001$; see Table 1). Capacitation-like acrosomal remodeling was most prevalent with zinc signatures 3 and 4 compared to zinc signatures 1 and 2 ($P < 0.0001$; 4 biological replicates; 10,000 spermatozoa analyzed per treatment). Acrosomal exocytosis occurred within the subpopulation of spermatozoa with zinc signature 4 and was greater than zinc signatures 1, 2, and 3 ($P < 0.001$; scale bar: 20 μm). **b** Zinc signature status corresponds with PI plasma membrane integrity in fresh (blue) and IVC spermatozoa (red). **c** As sperm plasma membrane integrity decreased, acrosomal remodeling and exocytosis occurred. *P*-values determined by the general linear model procedure in SAS 9.4

**Zinc signature associated with varied fertility in AI boars**. We examined possible individual variability in sperm zinc signature in AI boars with acceptable but varied fertility. In a small preliminary fertility trial ($n = 4$ boars with known fertility in AI service; fertility records in Supplementary Table 1), zinc signatures differed between high- and low-fertile boars both after IVC (Fig. 5a; original histograms located in Supplementary Fig. 4). Boars with high fertility have double the amount of signature 3 spermatozoa prevail after IVC (as percentage of population) as opposed to minimal signature 3 increase in low-fertility boars (Fig. 5b).

## Discussion

The inhibition of the zinc signature shifts may be related to the presence of Zn-binding UPS enzymes in spermatozoa, such as the RING-finger E3-type ubiquitin ligase UBR7[15]. The RING (really interesting new gene) finger ubiquitin ligases contain an amino acid motif that binds two zinc cations, allowing interaction with other proteins/enzymes, including establishment of a stable E3-substrate interaction required for protein ubiquitination. Other Zn-binding/Zn-containing proteins are likely present in spermatozoa since UBR7 is only detectable in the acrosomal region of fully differentiated spermatozoa and in the centriolar vault of the sperm tail-connecting piece. For example, sperm DNA-binding

**Table 1 Statistical analysis of zinc signature and acrosomal status**

|  | No remodeling | Remodeled | Exocytosed |
|---|---|---|---|
| Signature 1 | 93.0 ± 6.8%[Aa] | 4.0 ± 4.7%[Ba] | 3.0 ± 2.6%[Ba] |
| Signature 2 | 95.0 ± 2.6%[Aa] | 3.4 ± 2.9%[Ba] | 1.6 ± 1.2%[Ca] |
| Signature 3 | 11.1 ± 5.8%[Ab] | 81.0 ± 8.5%[Bb] | 7.9 ± 2.9%[Aa] |
| Signature 4 | 7.0 ± 9.9%[Ab] | 62.2 ± 12.9%[Bb] | 30.7 ± 3.0%[Cb] |

Data are presented as mean ± s.d. (three biological replicates). Values with different uppercase superscripts ([A,B,C,D]) indicate significant difference of the acrosomal status (P-value ≤0.0001) and lowercase superscripts ([a,b,c]) indicate significant difference of zinc signatures (P-value ≤0.0002) as determined by the GLM procedure in SAS 9.4. Both PI$^+$ and PI$^-$ cells were included in this analysis. A total of 10,000 cells were measured for each replicate

protamine PRM2 contains a zinc-finger domain[16], which plays a role in stabilization of sperm chromatin and inhibition of transcription. Protamines are believed essential for sperm head condensation and DNA stabilization during spermiogenesis, and alterations in sperm DNA protamination are associated with male infertility[17]. Further, JAMM (JAB1/MPN/Mov34 metalloenzyme) motif of the proteasomal regulatory subunit PSMD14/Rpn11 is a metalloprotease-like Zn site in the 26S proteasome[18]. Matrix metalloproteinases are zinc-dependent enzymes known mainly for their ability to digest extracellar matrix[19], with MMP-2 and MMP-9 being reported in human seminal plasma[20]. Inhibition of zinc-dependent metalloproteases hindered sperm passage through oocyte vestments during IVF[21]. Related to aforementioned protamine sperm chromatin stabilization, Zn-ion release from the nuclear zinc bridges described in human spermatozoa could facilitate sperm nucleus decondensation after fertilization, as a prelude to the formation of the zygotic paternal pronucleus[22,23]. Further studies thus should examine the relationship between sperm zinc signature and the aforementioned Zn-containing sperm proteins.

The changes seen in fresh, ejaculated zinc signature and the incubated, non-IVC spermatozoa could be heat-induced during the early stages of capacitation. Temperature influence on capacitation state could be related to the 35–38 °C thermotropic phase transition identified in the plasma membrane of ram spermatozoa[24]. These findings are important for livestock semen handling methods prior to AI.

Sperm capacitation, although required for fertility, is a terminal maturation event that leads to rapid cell death unless fertilization occurs[10]. The superimposition of zinc ion labeling and PI labeling in flow cytometric scatter plots allows us to subdivide spermatozoa within the boar ejaculate into four subpopulations (Fig. 3b). Thus, the disproportional representation of signatures 3 and 4, associated with sperm capacitated state and death, may indicate low-fertility ejaculates. Cell membrane changes heralded by PI incorporation in the sperm head at capacitation are concomitant with acrosomal remodeling signaled by lectin PNA binding, even though they occur at the opposite poles of the sperm head (Fig. 3c). As indicated in our time-lapse study, PI intensity changes over the course of capacitation and there are two subgroups of PI$^+$ spermatozoa: PI$^+$ live with plasma membrane change and PI$^+$ cell death (Fig. 2c). Although the signature 2 spermatozoa are the hyperactivated and ZP-interacting ones in Supplementary Movie 2, it is likely that a rapid transition occurs through these last stages of capacitation that cannot be distinguished within limitations of today's technology. After this rapid transition from zinc signature 2 to 3 and associated acrosomal changes, we hypothesize that the final cell death occurs rapidly if fertilization is unsuccessful. This might be the reason for sequential sperm capacitation observed within the oviductal sperm reservoir.

Comparison of zinc signature patterns in boars with varied fertility indicates potential of Zn probes in the evaluation of livestock sperm quality. While such findings with a small group of boars are preliminary, Zn fluorometry could be also given consideration in human andrology and infertility diagnostics. For instance, the sperm content of flagellar voltage-gated proton channel HVCN1 varies between human donors[3]. The HVCN1 regulates human CatSper Ca$^{2+}$ channel localized at the flagellum[3]. With CatSper activation required for hyperactivation and ultimately male fertility[25], it seems reasonable that zinc signature changes are reflective of this biological event and necessary for preparing the spermatozoa for hyperactivation. Other studies have shown that spermatozoa with higher fertilizing ability have increased changes in capacitation-induced biomarkers[26]. Altogether, this encourages dedicated trials with high statistical power sperm sample sets, aspiring to validate zinc signature as a candidate fertility marker. Such findings not only indicate the existence of sperm subpopulations capable/incapable of fertilizing the oocyte, but even more so that sequential capacitation and resulting waves of sperm release from the sperm reservoir, originally thought to be primarily driven by female reproductive tract-issued signals[27–29], are rather co-dependent of sperm subpopulation (Fig. 6a, b). Other recent studies hint at the significance of Zn$^{2+}$ for sperm structure and function/fertility. A decrease in Zn content of human seminal plasma has been associated with infertility stemming from accidental Chernobyl radiation in Ukraine[30] and individuals with high levels of asthenoteratozoospermic spermatozoa (low or no motility) have reduced seminal plasma Zn levels[31]. These findings are consistent with other studies reporting fertile males having increased seminal plasma Zn levels compared to infertile men[32,33]. Optimization of semen Zn$^{2+}$ and/or zinc containing protein(s) levels could thus improve the outcomes of AI in livestock and assisted reproductive therapy in humans.

The signature we describe here is likely representative of Zn$^{2+}$ being involved in multiple steps of sperm capacitation. At ejaculation, sperm motility is highly dependent upon ionic environment[34]. It is well understood that Zn$^{2+}$ can modulate cellular signaling, as well as protein kinase and phosphatase activities[35], and inhibit proteasomal activity[36]. Porcine seminal plasma reportedly contains 1.6–3.6 mM Zn$^{2+}$[37], the highest known levels of Zn$^{2+}$ found in any bodily fluid, thus likely to be serving some biological function. Further, Zn-binding seminal plasma proteins may protect the sperm plasma membrane against cold shock[38]. As mentioned earlier, Zn$^{2+}$ negatively regulates HVCN1 channel, the main proton extrusion mechanism in human spermatozoa[3]. Certain aspects of early capacitation events such as Ca$^{2+}$ influx require HVCN1 activation by Zn$^{2+}$ removal. The Zn$^{2+}$ content in the human sperm flagella is negatively correlated to sperm motility[39]; however, this may be due to capacitation-induced hyperactivation being inhibited in the presence of high Zn$^{2+}$. Further, Zn$^{2+}$-chelator DEDTC has been shown to immobilize spermatozoa, reaffirming the role of Zn$^{2+}$ in sperm motility[40]. Beyond sperm capacitation, the zinc signature or sperm zinc signaling of spermatozoa bound to the oocyte ZP could be altered by the Zn spark[1,2] triggered by the first fertilizing spermatozoon as well be altered by the 300% increase of zinc content in the ZP matrix following the Zn spark[41]. Altogether, such massive release of extracellular Zn$^{2+}$ and increased ZP Zn content could establish a combined zinc shield. Consequently, this zinc shield could derail Zn signaling in the spermatozoa surrounding the oocyte as an added barrier to polyspermic fertilization. Such mechanism seems plausible in consideration that Zn has been shown to be chemorepulsive to fertilization-competent human, mouse, and rabbit spermatozoa[42]. Furthermore, acrosin and matrix metalloproteinase MMP2, two of the proteinases implicated in sperm-ZP

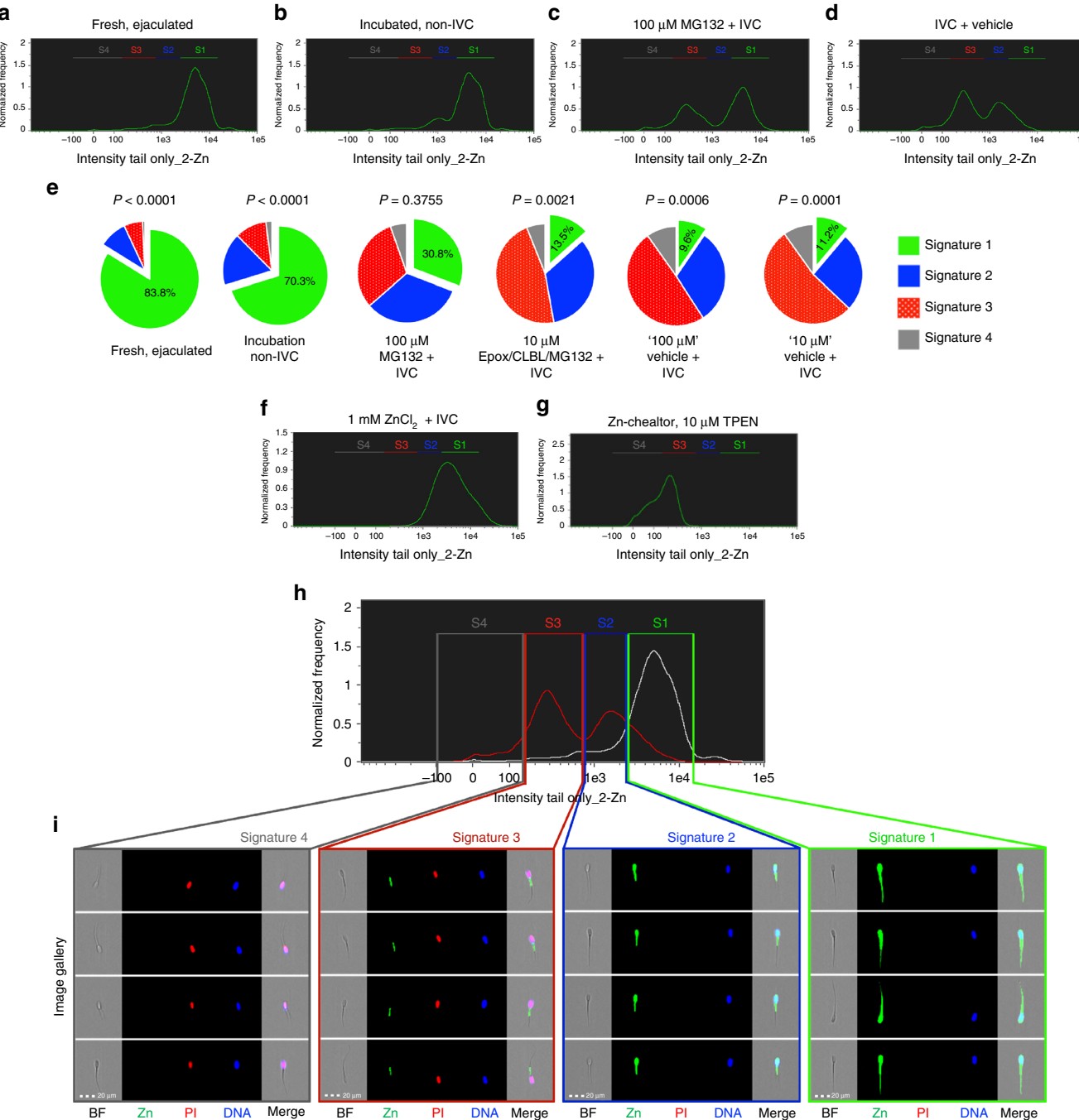

**Fig. 4** Modulation of internal and external $Zn^{2+}$ during IVC. **a** Fresh, ejaculated spermatozoa have mostly signature 1. **b** After 4 h under non-IVC conditions, few spermatozoa underwent spontaneous early-stage capacitation. **c** Proteasomal inhibitor MG132 (100 μM) prevented some of the IVC-induced zinc signature changes compared to **d** (IVC with MG132-vehicle); see Supplementary Fig. 2g for no vehicle IVC control. **e** Ten-micrometer Zn-chelator TPEN altered the zinc signature; see Supplementary Fig. 2h for TPEN vehicle treatment (P-value by treatment across replicates). **f** Addition of 1 mM $ZnCl_2$ + IVC prevented IVC-induced zinc signature changes. **g** Pie chart illustration of select IVC treatments (green: signature 1; blue: signature 2; red: signature 3; gray: signature 4). Treatment P-values are shown in Table 2 as determined by the general linear model procedure in SAS 9.4. 10,000 sperm per sample analyzed. **h** Histograms of non-capacitated (non-IVC, white) and capacitated (IVC, red) sperm populations from IBFC analysis **i** Images from IBFC gallery representing individual zinc signatures (scale bar: 20 μm). Each spermatozoon analyzed has the following images acquired: bright field (BF); $Zn^{2+}$ reporting probe FZ3 (Zn); sperm viability/plasma membrane integrity probe propidium iodide (PI); live DNA stain Hoechst 33342 (DNA); and side scatter (not displayed), with a merger of the four images (Merge)

penetration are present in the inner acrosomal membrane[43], (the location believed to be the leading edge of the sperm head during ZP penetration), have been shown to have their activities reduced and/or inhibited by Zn (human and bovine sperm acrosin[44], paddlefish acrosin-like activity[45], and brain MMP2 activity[46]).

Increased $Zn^{2+}$ concentration in bovine IVF media has previously been shown to inhibit fertilization[47] adding further support to the effective role of the zinc shield. Based on the present data, sperm zinc signature likely changes as the spermatozoa advance through the female reproductive tract and progress

**Table 2 Effect of proteasomal inhibition on zinc signature**

| Treatment | Signature 1 | Signature 2 | Signature 3 | Signature 4 | P-value |
|---|---|---|---|---|---|
| Fresh, ejaculated | 83.8 ± 1.8%[Aa] | 9.2 ± 0.9%[b] | 6.1 ± 1.2%[Ab] | 0.9 ± 0.4%[Ac] | P < 0.0001 |
| Incubation, non-IVC | 70.3 ± 2.5%[Aa] | 17.3 ± 3.0%[b] | 10.4 ± 2.7%[Ab] | 2.0 ± 2.7%[Ac] | P < 0.0001 |
| 100 μM MG132 + IVC | 30.8 ± 13.1%[B] | 32.9 ± 19.5% | 31.0 ± 4.9%[B] | 5.4 ± 4.9%[AB] | P = 0.3755 |
| 10 μM Epox, CLBL, MG132 + IVC | 13.5 ± 4.0%[BC] | 33.7 ± 8.8% | 46.8 ± 3.6%[C] | 6.1 ± 3.6%[B] | P = 0.0021 |
| "100 μM" Vehicle + IVC | 9.6 ± 3.3%[Ca] | 31.3 ± 7.0%[b] | 49.4 ± 4.5%[Cc] | 9.8 ± 4.5%[Ba] | P = 0.0006 |
| "10 μM" Vehicle + IVC | 11.2 ± 2.5%[Ca] | 25.8 ± 3.6%[b] | 53.1 ± 3.8%[Cc] | 9.9 ± 3.8%[Ba] | P = 0.0001 |
| P-value | P < 0.0001 | P = 0.4119 | P < 0.0001 | P < 0.0003 | |

Data are presented as mean ± s.e.m. (three biological replicates). Values with different uppercase superscripts ([A,B,C,D]) indicate significant difference between the control (fresh, ejaculated spermatozoa and vehicle controls) and treatment groups and lowercase superscripts ([a,b,c]) indicate significant difference between signatures as determined by Duncan's multiple range test (P-value by treatment and signature in table). Both PI[+] and PI[−] cells were included in this analysis. Treatment column refers to proteasomal inhibitors MG132, clasto-lactacystin ß-Lactone (CLBL), and epoxomicin (Epox.). A total of 10,000 cells were measured for each data point

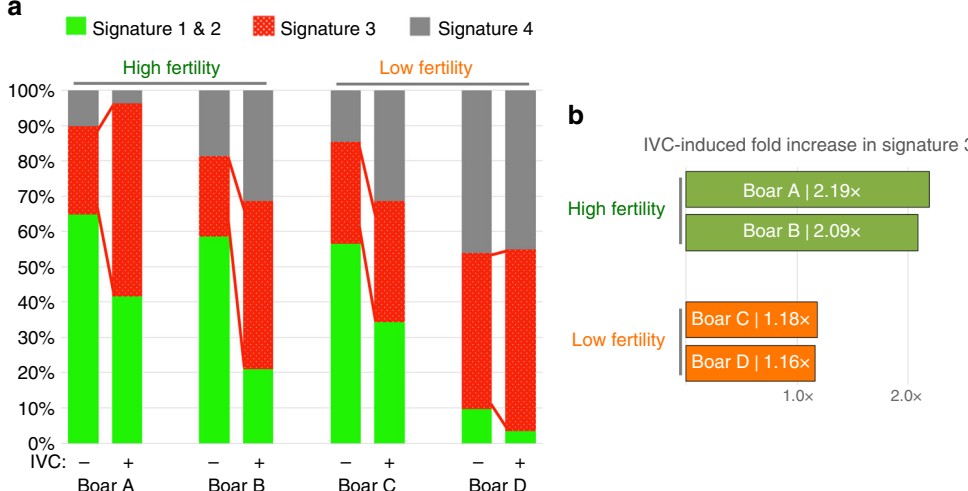

**Fig. 5** Zinc signature associated with varied fertility in AI boars. **a** Zinc signature in four boars of known high or low fertility, before and after IVC. **b** High-fertility boars had double the amount of spermatozoa with signature 3 after IVC compared to minimal increase in low-fertility boars

through different stages of capacitation; the proposed, reciprocal sperm, and oocyte $Zn^{2+}$ signaling for the blockage of polyspermy is outlined in Fig. 6b. Lastly, porcine ZP glycoproteins are highly glycosylated with neutral and acidic N- and O- linked oligosaccharides. N-glycans released from the ZP3-glycoprotein mixture are composed of neutral and acidic structures in approximate molar ratio of 1:2[48]. While neutral zona N-glycans have been found to be the determining sugar signals in the pig[49,50], only little is known about the function of acidic moieties of ZP glycoproteins[51]. It is plausible that $Zn^{2+}$, after $Zn^{2+}$ efflux associated with gamete fusion, may electrostatically bind to sialic residues of acidic oligosaccharides of ZP, altering ZP polarity and thus act as a fast block of polyspermy on the egg coat level by creating a zinc shield at the time of sperm-oolemma fusion. A similar effect could be achieved by $Zn^{2+}$ on the sperm surface, but most probably via a different mechanism, such as conformational change of ZP receptors resulting in decreased sperm affinity for the ZP.

Such findings shift the paradigm of anti-polyspermy defense mechanisms and complement the discovery of the oocyte zinc spark. Herein, if capacitation is defined solely as the capacity to fertilize, it would seem reasonable that the oocyte can decapacitate spermatozoa to some degree by manipulating sperm Zn levels; however, direct relationship with zinc signature should be explored.

In conclusion, the novel biological phenomenon of the sperm zinc signature is an early indicator of sperm capacitation and a candidate biomarker of sperm quality/fertility. The sperm

proteasome, at least in the subpopulation of capacitation competent spermatozoa, co-regulates earlier events of sperm capacitation than originally realized, and the Zn-chelator TPEN quenches the sperm zinc signature. The study of sperm zinc signature may further enhance our understanding of fertilization, including but not limited to the oocyte zinc shield formed upon fertilization to prevent polyspermy and sperm subpopulations capable of fertilization.

## Methods

**Reagents**. All reagents unless otherwise noted were from Sigma. FluoZin™-3, AM (FZ3; zinc probe) from ThermoFisher (F24195) was reconstituted with DMSO to a stock solution of 500 μM. Lectin PNA (A. hypogea/peanut agglutinin) conjugated to Alexa Fluor™ 647 (PNA-AF647) was from Invitrogen™ (L32460). Fluo-4 NW (calcium probe) from ThermoFisher (F36206) was reconstituted using kit provided assay buffer. Hoechst 33342 (H33342) from Calbiochem (382065) was reconstituted with $H_2O$ to a stock solution of 18 mM. PI from Acros Organics (AC440300010) was reconstituted with $H_2O$ to a stock solution of 1 mg mL$^{-1}$. Proteasomal inhibitors were from Enzo Life Sciences: MG132 (BML-PI102) was reconstituted with DMSO to a stock solution of 20 mM; epoxomicin (Epox, BML-PI127) was reconstituted to a stock solution of 20 mM (using MG132 stock); and clasto-lactacystin β-Lactone (CLBL, BML-PI108) was reconstituted with DMSO to a stock solution of 5 mM. Zn-chelator TPEN from Tocris (16858-02-9) was resuspended with 1:100 EtOH:$H_2O$ to a stock solution of 1 mM. Bovine serum albumin was from Sigma (A4503). Anti-phosphotyrosine antibody, clone 4G10® was from EMD Millipore (05-321).

**Semen collection and processing**. Boar semen collection was performed under the guidance of approved Animal Care and Use (ACUC) protocols of the University of Missouri-Columbia. Boar semen for the fertility trial was collected, extended, and shipped by overnight parcel from a private boar stud following their

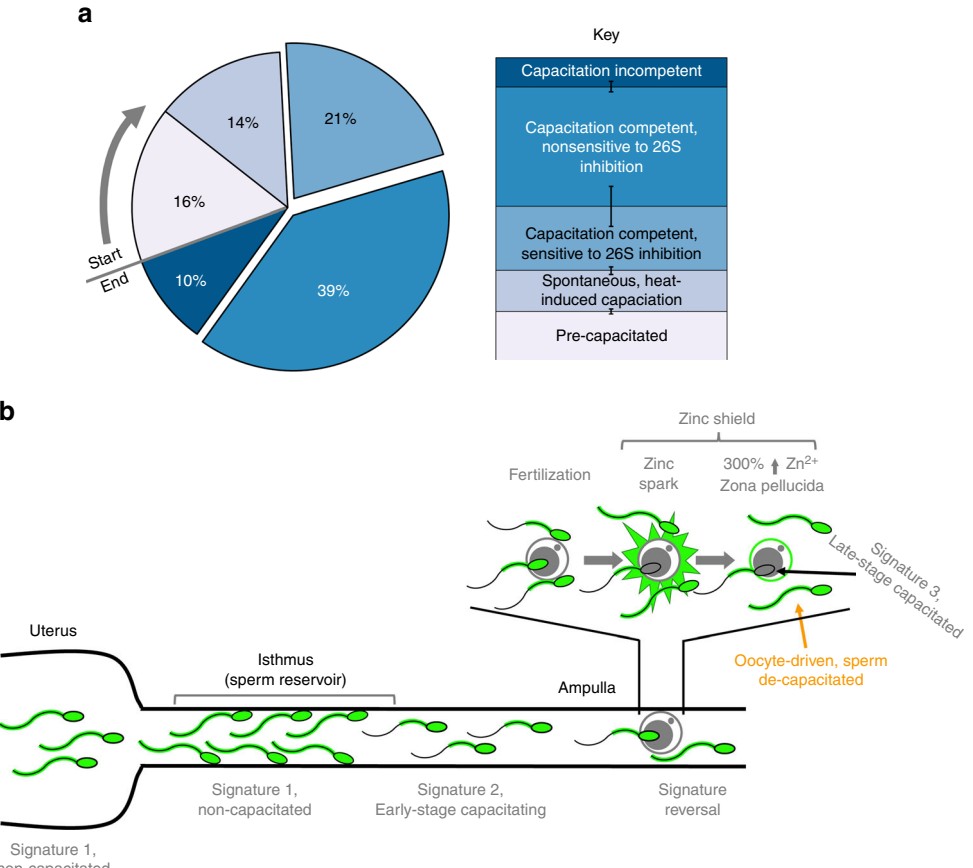

**Fig. 6** Proposed zinc signature population interpretation. **a** Interpretation of zinc signature meaning and population segregation: 16% of fresh, ejaculated spermatozoa had undergone early-stage capacitation upon semen collection (lightest blue working to darkest); 14% of spermatozoa spontaneously undergo early-stage capacitation during incubation without IVC inducers; 60% of spermatozoa remained capacitation competent with IVC inducers, with 21% sensitive to proteasomal inhibition; remaining 10% of sperm were capacitation incompetent under IVC conditions (darkest blue) (s.e. bars included). **b** Proposed zinc signature changes throughout female reproductive tract and oocyte zinc spark interference with sperm zinc signature as a combined polyspermy defense mechanism, the zinc shield

established standard operating procedures and was not blinded. Boar collection was performed using standard two gloved hand technique[9]. Only ejaculates with >80% motility were used and no randomization was necessary as only one boar was studied at a time. The sperm-rich fraction of the boar ejaculate was used, except all IVC proteasomal inhibition studies used the pre-sperm-rich fraction, which had increased viability and sensitivity to 26S inhibition. Semen was immediately extended, within 2 °C, five times in BTS semen extender. Sperm concentration was then determined using a hemocytometer. All washes were performed with a swing hinge rotor centrifuge at 110×$g$ for 5 min. The number of washes and g-force used were minimized as these were found to compromise results. Frozen-thawed bull spermatozoa were processed similarly as boar spermatozoa after being thawed for 45 s in a 35 °C water bath. For human spermatozoa, sperm donors signed informed consent and the samples were coded as to make the donors unidentifiable to researchers. All human sperm samples were handled and processed strictly as stipulated by an approved Internal Review Board (MU IRB) protocol. Donors were recruited by placing an advertisement for new fathers in the university mass e-mail newsletter. All semen were collected onsite at the Missouri Center for Reproductive Medicine and Fertility clinic. Samples were then transported to the laboratory for analysis.

**In vitro capacitation**. Fresh boar spermatozoa were capacitated using a protocol that rendered them capable of recognizing and binding to ZP, as well as undergoing acrosomal exocytosis and penetrating the oocyte ZP[11]. IVC-induced protein tyrosine phosphorylation changes are shown in Supplementary Fig. 5; acrosomal status and plasma membrane changes are shown in Fig. 2. Briefly, spermatozoa were washed of seminal plasma once with noncapacitating media (NCM), a modified TL-HEPES medium, free of calcium dichloride ($CaCl_2$) and addition of 11 mM D-glucose, with pH adjusted to 7.2. Spermatozoa were then resuspended in 0.5 mL IVC media, TL-HEPES-PVA supplemented with 5 mM sodium pyruvate, 11 mM D-glucose, 2 mM $CaCl_2$, 2 mM sodium bicarbonate, and 2% (m/v) bovine serum albumin, and incubated in a 37 °C water bath for 4 h, with eppendorf tube rotation

performed every 60 min. Control incubations under non-IVC conditions used NCM. Proteasome inhibitors (100 μM MG-132 and 10 μM Epox/CLBL/MG-132) were mixed with IVC media prior to sperm pellet resuspension. A 100 μM MG-132 and "100 μM" vehicle contained 0.5% (v/v) DMSO. A 10 μM Epox/CLBL/MG-132 and "10 μM" vehicle contained 0.3% (v/v) DMSO. PVA helped to reduce sperm aggregation and spermatozoa were pipetted repeatedly to dissociate sperm aggregates in a satisfactory manner prior to IBFC data acquisition. To confirm normal capacitation in our experimental IVC media, compared to 15 mM sodium bicarbonate IVC media, we refer readers to Supplementary Movie 1 (hyperactivation status) and Supplementary Fig. 5 (tyrosine phosphorylation status). Both media support hyperactivated motility. Unlike murine or rodent sperm tyrosine phosphorylation, porcine tyrosine phosphorylation is much more modest, with less prominent changes during the course of capacitation. Therefore, new bands after capacitation appear only at the molar weights of 32 kDa (acrosin-binding protein), and 21 kDa protein (phospholipid hydroperoxide glutathione peroxidase). Our results are in accordance with previous studies[52–54]. Final acrosome and plasma membrane modification status is similar at the end of IVC regardless of the two IVC treatment conditions; however, the rate of change and cell death is faster in 15 mM sodium bicarbonate-containing medium than experimental IVC medium (Fig. 2a–d). Altogether, this supports the use of experimental IVC medium over 15 mM sodium bicarbonate-containing medium to display the prolonged lifespan of spermatozoa as seen in in vivo capacitation.

**TPEN Zn chelation**. $Zn^{2+}$ chelation was performed using TPEN (membrane permeable). Ten micromolar TPEN was incubated with 40 million sperm per mL for 1 h. Stock TPEN: 1 mM in 1:100 EtOH:$H_2O$.

**Multiplex fluorescence probing**. Upon 4 h of IVC, a sample size of 100 μL (4 million spermatozoa) were incubated 30 min with 1:200 H33342, 1:200 PI, and 1:100 FZ3 for epifluorescence microscopy. Lower probe concentrations were necessary for IBFC due to camera detection differences, thus 1:1000, 1:1000, and

1:500 were used, respectively, with inclusion of 1:1000 PNA-AF647. For Fluo-4 calcium probe, we followed manufacturer protocol using identical cell concentrations. Spermatozoa were then washed of probes once and resuspended in corresponding IVC treatment media to allow complete de-esterification of intracellular AM esters, as suggested by ThermoFisher's FZ3 protocol, followed by an additional wash and resuspended in 100 μL PBS for IBFC analysis (or added to a slide for epifluorescence microscopy imaging).

**Epifluorescence microscopy imaging**. Live spermatozoa were imaged using a Nikon Eclipse 800 microscope (Nikon Instruments Inc.) with Cool Snap camera (Roper Scientific, Tuscon, AZ, USA) and MetaMorph software (Universal Imaging Corp., Downington, PA, USA). Images were adjusted for contrast and brightness in Adobe Photoshop CS5 (Adobe Systems, Mountain View, CA) to match the fluorescence intensities viewed through the microscope eyepieces.

**Image-based flow cytometric data acquisition**. IBFC data acquisition was performed following previous methodology[55]. Specifically, using a FlowSight flow cytometer (FS) fitted with a ×20 microscope objective (numerical aperture of 0.9) with an imaging rate up to 2000 events per s. The sheath fluid was PBS (without $Ca^{2+}$ or $Mg^{2+}$). The flow-core diameter and speed was 10 μm and 66 mm per s, respectively. Raw image data were acquired using INSPIRE® software. To produce the highest resolution, the camera setting was at 1.0 μm per pixel of the charged-coupled device. In INSPIRE® FS data acquisition software, two bright-field channels were collected (channels 1 and 9), one FZ3 image (channel 2), one PI image (channel 5), one side scatter (SSC, channel 6), one H33342 (channel 7), and one PNA-AF647 image (channel 11), with a minimum of 10,000 spermatozoa collected. The following lasers and power settings were used: 405 nm (to excite H33342): 10 mW; 488 nm (to excite FZ3): 60 mW; 561 nm (to excite PI): 40 mW, 642 nm (to excite PNA-AF647): 25 mW; and 785 nM SSC laser: 10 mW.

**IBFC data analysis**. Data were analyzed using IDEAS® analysis software from AMNIS EMD Millipore. Gating approach used standard focus and single cell gating calculations created by IDEAS software (Supplementary Fig. 1a, b). To further clean up data for analysis, Feature Finder function was used to discover image-based calculations to discard spermatozoa laterally aligned with the camera, as opposed to anteriorly/posteriorly aligned (Supplementary Fig. 1c). Traditional flow cytometric analysis methods does not allow to distinguish signature 1 and 2 based on whole-cell FZ3 intensity, therefore creating a mask that only analyzes the sperm tail proved to be key in distinguishing these two populations. Such mask was created by taking a morphology mask of the bright field (Supplementary Fig. 1d), subtracting a 4-pixel dilation of H33342 (Supplementary Fig. 1e), resulting in a mask to analyze fluorescence in the tail region only (Supplementary Fig. 1f, g). Mask dilation of H33342 fluorescence was necessary because H33342 labeling of the sperm nucleus did not cover the entirety of the sperm head, where FZ3 signal was high. This combined gating and masking strategy provided robust clean data, ready for signature analysis by plotting FZ3 intensity of this masked region, which is impossible with traditional flow cytometry. Gating boundaries for signatures 1, 2, and 3 were determined by the population segregations of fresh/ejaculate and IVC + vehicle treatments with signature status confirmed in the image gallery. Gating boundaries between signatures 3 and 4 were less evident in histograms and placed where spermatozoa lost FZ3 signal as determined using the image gallery.

**Sequential sperm extraction treatment**. Approximately 200 million washed spermatozoa were used per single treatment, which was conducted by adding 100 μL of a relevant reagent; proteinase and phosphatase inhibitors included. In the first step, PBS was added to the sperm pellet, allowed to incubate on ice for 30 min and spun. In second step, the pellet was reused and 0.75 M KCl in PBS was added and incubated on ice for 30 min and spun down. The pellet was washed once with PBS to remove residual salt and reused in the third step for treatment with 30 mM n-octyl-β-D-glucopyranoside (OBG) in PBS. The sperm after each treatment step were analyzed for their zinc signature.

**Western blotting**. Sperm pellets (15 million spermatozoa per pellet) were mixed with reducing SDS-PAGE loading buffer, boiled for 5 min and briefly spun at 5000×g. The SDS-PAGE was carried out on a 4–20% gradient gels (PAGEr Precast gels; Lonza Rockland, Rockland, ME, USA) as previously described[13]. The molecular masses of the separated proteins were estimated by using prestained Prosieve protein colored markers (Lonza Rockland) run in parallel. After SDS-PAGE, proteins were electro-transferred onto a PVDF Immobilon Transfer Membrane (Millipore, Bedford, MA, USA) using an Owl wet transfer system (Fisher Scientific) at a constant 50 V for 4 h for immunodetection[13].

**Statistics**. All results are presented as mean ± s.e. SAS 9.4 GLM procedure and Duncan's multiple range test was used to analyze the replicates. Bartlett and Leven tests found the sample set to be homogenous.

**Animal models**. Domestic (Sus scrofa) boars 2 years of age were used for all experiments performed. Domestic (Bos taurus) bulls 3 years of age were used to confirm zinc signature presence.

**Data availability**. There are no database deposited data in this article. All relevant data are available from the authors.

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

## Acknowledgements

Supported by the National Institute of Food and Agriculture (NIFA), U.S. Department of Agriculture (USDA) grant number 2015-67015-23231 (P.S.), USDA NIFA Graduate Fellowship award number 2017-67011-26023 (K.K.), grant number 5 R01 HD084353-02 from NIH National Institute of Child and Human Development (P.S.), European Regional Development Fund (ERDF) BIOCEV grant CZ.1.05/1.1.00/02.0109 (M.Z.), the Czech Academy of Sciences (RVO:866525036) (M.Z.) and seed funding from the Food for the 21st Century Program of the University of Missouri (P.S.). We thank staff of the National Swine Research Resource Center, University of Missouri, as well as Dr. Randall Prather and his associates for assisting in boar semen collection and processing. We also thank Mr. Dalen Zuidema for providing MII porcine oocytes, Ms. Lauren Mayo for assistance with statistical analysis, and Ms. Katherine Craighead for clerical support.

## Author contributions

K.K. and P.S. designed the research; K.K., M.Z., E.Z.D., M.S., and P.S. performed the research; K.K., M.Z., and P.S. analyzed the data; K.K., M.Z., E.Z.D., and P.S. wrote the paper.

## Additional information

**Competing interests:** The authors declare no competing interests.

