## [Peer Review File · Nature Communications]

Reviewers' comments:

Reviewer #1 (Remarks to the Author):

In this manuscript, the authors describe the use of a fluorescent Zn²⁺ indicator to reveal Zn²⁺ distribution in boar sperm before and after incubation under capacitating conditions (sperm capacitation being defined as gaining the ability to fertilize oocytes). They classified Zn²⁺ distribution into 4 patterns and then associated a change in the expression of the patterns with the presumed capacitation of the sperm. They also examined effects of proteasome inhibitors, extracellular Zn²⁺, and Zn²⁺ chelators on Zn²⁺ distribution patterns. They claim that this work ties changes in Zn²⁺ distribution in sperm to the process of capacitation, and that the patterns of Zn²⁺ distribution represent a novel contribution to the understanding of sperm capacitation. Unfortunately, the state of capacitation was only assumed based on sperm treatment and not assayed directly.

The authors have not built a clear case for how their findings advance knowledge in the field of mammalian sperm biology; nor have they explained their findings in a way that would be understood and appreciated by a broader audience.

This manuscript requires extensive editing for clarity and accuracy. The necessary editing includes (but is not limited to) removing jargon and defining field-specific terms. I have listed below several specific concerns and examples of sentences needing revision, but my list is not comprehensive.

I do not understand the meaning of the title. What is unique about the "Zn²⁺ signature" in sperm?

Please remove all literature citations from the abstract.

Lines 17-19: Please clarify the meaning of this sentence. What exactly has never been described?

Lines 27-28: It is not clear which "paradigms" and "dogma" are being referred to.

Line 35: Please define "zinc spark".

Line 49: By "vitelline coat" do you mean "zona pellucida"?

Line 50: Please clarify this definition of propidium iodide.

Line 54: Please delete "unknowingly".

Lines 61-63: Please define clearly here how you objectively and quantitatively distinguished high Zn²⁺ levels from medium Zn²⁺ levels. Were fluorescence signals calibrated?

Line 64: Delete "/mitochondrial sheath".

Line 65: Did propidium iodide (PI) labeling confirm that "signature 4" represents dead sperm? Were there any dead sperm (PI positive) assigned to the other Zn²⁺ signature categories? What percentage of sperm showing signature 3 pattern were dead? Please clarify whether dead (PI-positive) sperm were excluded from the analysis shown in Table 1.

Lines 70-73: How can it be concluded that a decrease in fluorescence meant that sperm underwent "temperature-induced capacitation"? Capacitation was not assayed.

Line 74: Since proteasome activity was not measured, it would be more accurate to state here that a higher portion of sperm retained signature 1 when proteasome inhibitors XXX were added to IVC

medium. Include a citation here for the established effectiveness of these inhibitors on sperm proteasome activity.

Line 77: Please define "TPEN", using the correct chemical term.

Lines 78-80: Please clarify what you mean by "outer structures" of sperm. The reader could go to extended Figure 3 to see how the sperm were treated; however, that should not be necessary and also would not explain what is meant by "outer structures".

Lines 103-104: Reference #17 is not appropriate here, because it does not support this statement.

Lines 104-107: I do not understand this sentence.

Lines 118-121: To my knowledge, there is no solid basis for this conclusion. Also, a reference must be provided for the statement about female reproductive tract signals driving sperm release from the oviductal storage reservoir.

Line 153: Please delete the word "pathological".

Line 165: How is the Zn²⁺ "signature" a "novel biological phenomenon"? It is simply an indicate of the distribution of Zn²⁺ in sperm.

Table 1: The numbers of cells per sample should be given. Two replicates do not seem to be an adequate amount of sampling. Define the control groups mentioned in Line 294. Please define MG132, CLBL, and Epox in the caption. Please clarify whether dead (PI positive) sperm were included in these data.

Fig 1. Some of the labeling of 1f and 1g is too small; the images in 1g are too small.

Line 225: what is meant by "to let probes rest"?

Line 233: Unless I misunderstood this sentence, I understood that the brightness of each fluorescence image was adjusted for brightness and contrast to match what was seen through the microscope. It would not be acceptable to adjust images subjectively and individually.

Line 252: Data were.

Line 256: Please define what is meant by "laterally aligned". Lines 256-259: Why were sperm that were "laterally aligned" eliminated from analysis if you only used tail FZ3 fluorescence to distinguish signature 1 from signature 2?

Line 289: Provide the Latin name for the species of pig, the name of the breed, and the number of boars used.

Reviewer #2 (Remarks to the Author):

This manuscript describes on the loss of Zn²⁺ ions in capacitating pig sperm using FluoZin TM -3, AM staining and fluorescent imaging flow cytometry. The relative proportion of sperm that show the highest loss of Zn²⁺ was higher in lower fertile boars when compared to higher fertile boars. Although this is a novel finding with potential importance in the understanding how the sperm cell becomes prepared to fertilize the oocyte a number of points deserve attention.

1. The authors incubated sperm with 2 mM sodium bicarbonate which is a too low concentration for

inducing bicarbonate mediated events required for in vitro capacitation. Most capacitation media make use of >15 mM bicarbonate to allow an increase in intracellular cAMP levels. In porcine IVF protocols either a similar high bicarbonate levels is used to enhance intracellular cAMP, or alternatively, at 2 mM bicarbonate levels phosphodiesterase inhibitors are added to induce a similar enhancement of intracellular cAMP levels in sperm (caffeine is most commonly used). However, the here used low bicarbonate levels in combination with the absence of phosphodiesterase inhibitors make it questionable whether or not sperm actually undergo a full cAMP dependent protein kinase activation.

2. Replace "Zn²⁺ signature" into "Zn²⁺ distribution pattern" as signature is in my view vague terminology.

3. The authors have selected live cells for detecting Zn²⁺ fluorescent distribution patterns by gating out deteriorated cells (conform figure 2 A) on basis of PI staining. It would be good to include the pattern of deteriorated (PI+) cells. For instance do such PI+ cells have a S4 pattern?

4. It is known from different species including the pig that sperm capacitation induces to a varying extend sperm aggregation. Obviously, this was also observed in the present study (see in the Suppl Information Figure 1 B which depicts multi-sperm events). Although more than one sperm is in the image it is still possible to make a micrograph of all sperm in such an event and detect Zn²⁺ staining patterns in all individual sperm in such micrograph. This information now missing but it is well possible that in such sperm the capacitation specific Zn²⁺ distribution pattern is more frequently present when compared to the single sperm events.

5. The orientation of sperm is also of interest for the interpretation of sperm Zn²⁺ distribution patterns: The supplementary Figure 1 shows elegantly how out-of-focus correction, lateral positioning alignment and masking steps are done before data analysis. However, as shown in Suppl Fig 1 d-g and in Figure 1 g (S3 and S1) sperm go through the detecting spot either with their head on top or at the back. This will influence the distribution of light emitted from the passing sperm. It is not clear to me how the authors corrected for this head-first versus head-last orientation of sperm.

6. The changes in relative proportions of S1-S4 Zn²⁺ distribution patterns have not been correlated to other sperm capacitation responses in this manuscript. I wonder how these changes correlate with the induction of tyrosine phosphorylation, to hyper-activated sperm motility patterns and to the degree of artificial induction of acrosome reaction (i.e. after a Ca²⁺ ionophore or a lysoPC challenge). All these responses are considered to be signs of sperm capacitation.

7. The zinc shield idea postulated in figure 3 D (i.e. events happening to sperm termed as signature reversal Zinc spark) are highly speculative and not based on any experimental evidence.

8. Likewise this manuscript carries a style of scientific writing allowing too much premature but conclusive statements and trying to explain too many aspects of sperm capacitation/fertilization on only a limited amount of data.

These issues need to be taken care of in detail by the authors. Having said that, the authors have introduced elegantly the new technique of imaged based flow cytometry and their gating and masking strategies implemented are state-of-art and clearly the way to go for studies as presented here.

Reviewer #3 (Remarks to the Author):

General Comments: In this study, Kerns et al., report four types of sub-cellular Zinc (Zn²⁺) distributions in ejaculated porcine sperm using image-based flow cytometry (IBFC). The authors show that these "Zn²⁺ signatures" change during capacitation and can be regulated by proteasome activity and Zn²⁺ concentration. The authors suggest that a specific Zn²⁺ signature in porcine sperm is associated with sperm fertility.

The authors made a good use of IBFC to analyze sperm heterogeneity and report an interesting observation that porcine sperm cells are comprised of distinct subpopulations that can be classified

by distinct novel Zn²⁺ signatures. The approach enables to quantify many parameters at single cell levels, thus providing new insights on Zinc ion in sperm physiology. However, the experimental results to support the main conclusions (sperm cells with a specific zinc signature indicates higher fertility) remain very preliminary and speculative without clear molecular mechanisms or enough number of samples or replicates in the experiments. In addition, certainly there are areas the authors need to be more clear in their writing. The manuscript also requires more attention to details. For example, there are mislabel and order in the figure or in the figure legend (see the specific comments below).

Major Comments:

(1) General writing style: Overall, whole manuscript is written very vaguely. For example, in the Abstract, what is "zinc signature"? what is "new paradigm" and what is "older dogma"? The current abstract does not deliver what was done in the study without reading the whole manuscript.

(2) The authors generalize their observation as "mammal" in this manuscript. However, this needs to be more careful. Although Zn inhibits Hv1, the channel does not exist in rodent sperm, for example. This indicates the zinc binding proteins to make Zn signature would be quite different among the species. Most of all, it is unclear and not demonstrated in the manuscript whether the patterned Zn signature change in sperm is conserved phenomenon in mammalian sperm or present specifically in porcine sperm.

(3) General experimental design: In this study, single time point for sperm incubation for 4h is used to see the changes in their Zn signature. How fast is the transition happening during capacitation? Time-laps investigation could help to elucidate the signature change and sperm capacitation in boar.

(4) Line 40-44: This sentence is simply not right. Ca²⁺ entry via CatSper is not a pre-requisite for P-Tyr. In fact, it is quite the opposite (Chung et al, Cell (2014), Navarrete et al, J Cell Physiol (2015), Navarrete et al, Sci Rep (2016)).

(5) Line 46: what are the "certain aspects"? Please be specific.

(6) Related to Figure 1. Although sperm cells incubated under the capacitating conditions changed the distribution of the distinct patterns of their Zn²⁺ imaging of sperm, there is no direct evidence that the increased fraction (Signature 3) during capacitation is positively correlated with (or causal) to sperm fertility. In particular, sperm cells classified to have signature 3 and 4 group are PI positive. Therefore, it is not clear whether the signature alteration directly reflects sperm capacitation status or is simply due to sperm death and or damage to membrane. To clarify this, it is suggested to directly compare acrosome status and hyperactivated motility with the patterns of Zn distribution after capacitation using different approach other than IBFC.

(7) Line 72-74: temperature-induced capacitation is over-interpretation. It could be spontaneous. You need room temperature incubation, non-IVC condition required to claim it. Related to Figure 2 and 3. The temperature change in the experiment is over 10 C° (Room temperature, 20 - 25 C° vs capacitation condition, 37 C°). However, the temperature difference between porcine epididymis (36 - 37 C°) and oviduct (38 - 39 C°) is only 2-3 C°. Despite the larger temperature difference than that in in vivo status, only around 10 % fraction was shifted from signature 1 to signature 2. And also, the incubation temperature for capacitation (37 C°) is already similar to that in epididymis (36 - 37 C°) where the capacitation is prevented. Therefore, the result should be interpreted again with considering environment during natural mating.

(8) Line 77-78: Related to Figure 2 and Extended figure 3. This reviewer believe that the experimental data do not support that majority of ZFN signal is gone signature 3 remain (Figure 2G). Why cell permeable TPEN cannot chelate mitochondrial Zn²⁺?

The authors performed sequential extraction of sperm to investigate how Zn signature is related to membrane structures. However, PBS washing itself induces severe loss of Zn from sperm, followed by 0.75 mM KCl. Is KCl concentration a typo? It is not clear how 0.75 mM KCl after 150 mM NaCl in PBS can tell the ionic effect and the following non-ionic detergent effect on the Zn signature in sperm. Also, in this part, the authors already suggested that simple diffusion could result in depletion of Zn in sperm by PBS washing. Then, why this simple diffusion does not occur to both capacitation and non-capacitation conditions. Other possibility such as cell viability of sperm after PBS washing be excluded in the experiment?

(9) Lines 465-467: 0.75 mM KCl treatment?

(10) Regarding Figure 3. While understanding the difficulty of doing experiments with big animal models, $n=4$ is simply not enough to suggest that Zn signature could be used for fertility diagnosis. There is even no difference in Boar B of High Fertility and Boar C of Low fertility in overall Zn signature before IVC. Also there must be a mislabeling in Figure 3b. Boar B should have a higher fold change but the number does not reflect it. Also it will be more straightforward to express b in fold change as Figure 3a is already expressed in percentage. To rectify this point, authors need to increase the number of boars, or do another experiment to investigate fertility, such as in vitro fertilization. Also, correlative analyses could help to support author's suggestion, at least statistic level. There is also no rationale provided why the authors start to combine signature 1 and 2 from Figure 3.

(11) Lines 114-115: This reviewer was not able to find the result corresponding to "Hv1 mRNA abundance has been strongly correlated with male infertility." from the reference the authors cited, not even from the supplementary data.

(12) Lines 277-278: Again, 0.75 mM KCl used or 0.75M (750 mM) KCl used? In Lines 465-467, KCl concentration was 0.75 mM.

Minor Comments:

(1) Line 65: Signature 4, death sperm dead sperm

(2) Lines 169-172: the sentence is incomplete – verb missing. "The study of zinc signature many further enhance our understanding of..."

(3) Related to Figure 2. Figure legends does not match to figure. Panel numbering should be fixed.

(4) Related to Extended figure 2 and 3. In manuscript, extended figure 3 proceeds extended figure 2. The order should be changed. And also, numbers of panels in legends for extended figure 3 should be fixed.

Reviewer #4 (Remarks to the Author):

This is an interesting study that explores whether zinc signature of mammalian spermatozoa changes during capacitation and whether such feature could be used as a diagnostic measure. The study provides interesting insight on the various roles of zinc in sperm physiology. Overall, study is well-done, but additional data are needed to justify the conclusions.

1. It is known that sperm cells do not capacitate uniformly, and the capacitated sperm population is usually heterogeneous and comprised between 10% to 60% of capacitated sperm cells per fraction. How do authors know that they look at the capacitated spermatozoa? Additional experiments are required to either co-stain spermatozoa with the marker of tyrosine phosphorylation and look at the zinc signature, assess their acrosome status or cherry pick spermatozoa that exhibit hyperactivation as was shown in Chung JJ, Shim SH, Everley RA, Gygi SP, Zhuang X, Clapham DE. Cell. 2014

2. How do authors exclude the fact that TPEN can also bind calcium (it is not strictly zinc-specific), and one of the reason they see strong staining in the acrosome, is because it is abundant in calcium?

RESPONSE TO REVIEW, MS NCOMMS-17-14258-T: Zinc Ion Flux during Mammalian Sperm Capacitation

We thank all reviewers for thoughtful comments, which we address in detail below and in the revised manuscript by extensive new data addition and re-editing.

Reviewer #1

Unfortunately, the state of capacitation was only assumed based on sperm treatment and not assayed directly.

REPLY: We added extensive data with capacitation assays, including tyrosine phosphorylation and acrosomal remodeling.

The authors have not built a clear case for how their findings advance knowledge in the field of mammalian sperm biology; nor have they explained their findings in a way that would be understood and appreciated by a broader audience. This manuscript requires extensive editing for clarity and accuracy.

REPLY: We have revised the manuscript to convey our message about the significance and novelty of our findings. With regard to line items, following line edits have been completed as requested by reviewer (please see marked up manuscript copy):

Line 35: Please define “zinc spark”.

Line 54: Please delete “unknowingly”.

Line 64: Delete “/mitochondrial sheath”.

Line 74: Since proteasome activity was not measured, it would be more accurate to state here that a higher portion of sperm retained signature 1 when proteasome inhibitors were added to IVC medium

Line 77: Please define “TPEN”, using the correct chemical term.

Please clarify whether dead (PI positive) sperm were included in these data – we explain that we did include dead and moribund (PI-positive) spermatozoa in the analysis.

Line 153: Please delete the word “pathological”.

Line 252: Data were.

Line 289: Provide the Latin name for the species of pig, the name of the breed, and the number of boars used.

I do not understand the meaning of the title. What is unique about the “Zn²⁺ signature” in sperm?

REPLY: Unique was used in sense of not previously reported, to our knowledge in any cell type. Regardless, we have changed the title in response to other reviewers’ comments: Zinc ion flux during mammalian sperm capacitation.

Please remove all literature citations from the abstract.

REPLY: Removed.

Lines 17-19: Please clarify the meaning of this sentence. What exactly has never been described?

REPLY: Abstract has been rewritten to *Nat. Commun.* format; previous version already

explained that the ion flux associated with sperm capacitation has never before been described.

Lines 27-28: It is not clear which “paradigms” and “dogma” are being referred to.

REPLY: We deleted the wording concerning “old dogmas”; we realize that there actually is no old dogma to challenge regarding the role of zinc in capacitation. We explain now specify that new paradigm refers to the role of zinc ions during capacitation and fertilization.

Line 49: By “vitelline coat” do you mean “zona pellucida”?

REPLY: Yes, we changed it to “zona pellucida”.

Line 50: Please clarify this definition of propidium iodide.

REPLY: We now specify that PI is taken up exclusively by dead and moribund cells due to compromised plasma membrane.

Lines 61-63: Please define clearly here how you objectively and quantitatively distinguished high Zn²⁺ levels from medium Zn²⁺ levels. REPLY: Fluorescence intensities of the four signatures were measured exactly by meticulously calibrated FlowSight instrument, as we now specify in this sentence and detail elsewhere in the manuscript. Were fluorescence signals calibrated?

Line 65: Did propidium iodide (PI) labeling confirm that “signature 4” represents dead sperm? Were there any dead sperm (PI positive) assigned to the other Zn²⁺ signature categories? What percentage of sperm showing signature 3 pattern were dead? Please clarify whether dead (PI-positive) sperm were excluded from the analysis shown in Table 1.

REPLY: PI-/± sperm numbers were included in Table 1 and Table 1 caption was edited accordingly. PI does not only signify dead sperm (dying, moribund spermatozoa also take up PI); it also reflect plasma membrane integrity and permeability which changes with capacitation. We added wording on signature 4 spermatozoa having compromised/remodeled plasma membrane and included Figure 2j showing only sperm with capacitation-induced acrosomal remodeling and exocytosis are PI+.

Lines 70-73: How can it be concluded that a decrease in fluorescence meant that sperm underwent “temperature-induced capacitation”? Capacitation was not assayed.

REPLY: We changed the wording to clarify “early stage capacitation,” and this can be concluded since it is known that Zn negatively regulates Hv1 proton channel, which when activated increases intracellular pH and allows Ca²⁺ entry via CatSper. If temperature-induction is in question, the “fresh, ejaculated” sample remained at room temperature while “incubation, non-IVC” was at 37 degrees C.

Line 74: Include a citation here for the established effectiveness of these inhibitors on sperm proteasome activity.

REPLY: Citations have been added of papers in which we used specific proteasomal

substrates to measure sperm proteasome enzymatic activities with and without said inhibitors.

Lines 78-80: Please clarify what you mean by “outer structures” of sperm. The reader could go to extended Figure 3 to see how the sperm were treated; however, that should not be necessary and also would not explain what is meant by “outer structures”.

REPLY: We changed the wording as follows: With the exception of the midpiece, zinc ions appeared to be associated with the sperm surface, as the stepwise extraction removed Zn^{2+} tracer fluorescence early in treatment procedure

Lines 103-104: Reference #17 is not appropriate here, because it does not support this statement.

REPLY: We include specific reference on this line.

Lines 104-107: I do not understand this sentence.

REPLY: We rewrote this sentence: The distribution of zinc ion and PI double-labeling in scatter plots allow us to subdivide spermatozoa within the boar ejaculate into four subpopulations (**Fig. 2a**). Thus, the disproportional representation of signatures 3 and 4, associated with sperm capacitated state and death, may indicate low fertility ejaculates.

Lines 118-121: To my knowledge, there is no solid basis for this conclusion. Also, a reference must be provided for the statement about female reproductive tract signals driving sperm release from the oviductal storage reservoir.

REPLY: We believe it is well established that female reproductive tract controls sperm release to synchronize it with ovulation. Several mechanisms have been proposed, including but not limited to periovulatory temperature increase in the oviduct, chemoattractant release from ovulated oocyte-cumulus complexes and signaling molecules within the released ovarian follicular fluid. We now cite pertinent review articles here.

Line 165: How is the Zn^{2+} “signature” a “novel biological phenomenon”? It is simply an indicator of the distribution of Zn^{2+} in sperm.

REPLY: Simply because it has never been described until now.

Table 1: The numbers of cells per sample should be given. Two replicates do not seem to be an adequate amount of sampling. Define the control groups mentioned in Line 294. Please define MG132, CLBL, and Epox in the caption.

REPLY: We have added definitions of control groups and inhibitors; additional replicates have been added.

Fig 1. Some of the labeling of 1f and 1g is too small; the images in 1g are too small.

REPLY: We re-edited the figure panels, changed fonts.

Line 225: what is meant by “to let probes rest”?

REPLY: Changed to “to allow complete de-esterification of intracellular AM esters, as suggested by ThermoFisher’s FZ3 protocol”

Line 233: Unless I misunderstood this sentence, I understood that the brightness of each fluorescence image was adjusted for brightness and contrast to match what was seen through the microscope. It would not be acceptable to adjust images subjectively and individually.

REPLY: Yes, you understood it correctly. Raw, unprocessed data are available if any concerns arise.

Line 256: Please define what is meant by “laterally aligned”. Lines 256-259: Why were sperm that were “laterally aligned” eliminated from analysis if you only used tail FZ3 fluorescence to distinguish signature 1 from signature 2?

REPLY: Focal calculations are based mostly on the sperm head, and sperm that are laterally aligned (i.e. lying flat against the acquisition plane) are more likely to have their tails out of focus (while the head is in focus), which affects the intensity reporting of FZ3 and therefore discarded from analysis to provide uniform clean results. The phrase “as opposed to anteriorly/posteriorly aligned” has been added to (previous) line 256 for clarification.

Reviewer #2

1. The authors incubated sperm with 2 mM sodium bicarbonate which is a too low concentration for inducing bicarbonate mediated events required for in vitro capacitation. Most capacitation media make use of >15 mM bicarbonate to allow an increase in intracellular cAMP levels. In porcine IVF protocols either a similar high bicarbonate levels is used to enhance intracellular cAMP, or alternatively, at 2 mM bicarbonate levels phosphodiesterase inhibitors are added to induce a similar enhancement of intracellular cAMP levels in sperm (caffeine is most commonly used). However, the here used low bicarbonate levels in combination with the absence of phosphodiesterase inhibitors make it questionable whether or not sperm actually undergo a full cAMP dependent protein kinase activation.

REPLY: Thank you for noticing this and an opportunity to explain. A previous study done in our lab (Zimmerman et al., 2011) has shown this method capacitates sperm and allows for zona pellucida recognition, a signature of capacitation. Our media includes increased sodium pyruvate (5 mM), which has been shown to increase capacitation (hyperactivation and increased tyrosine phosphorylation) by increasing ATP (Hereng et al., 2011). We have included a new, confirmatory Western blot (SD Figure 4) to show that that there is no difference in the number and densities of the phosphotyrosine bands between our previously proven capacitation media and a medium with 15 mM sodium bicarbonate.

2. Replace “Zn²⁺ signature” into “Zn²⁺ distribution pattern” as signature is in my view vague terminology.

REPLY: We appreciate your concern. With your agreement, we would like to define the word “signature” early in the manuscript (in abstract) as a “Zn²⁺ distribution pattern” and then continue using “signature” in text and figures for the sake of brevity. Also, we

have removed “signature” from the title of paper accordingly.

3. The authors have selected life cells for detecting Zn²⁺ fluorescent distribution patterns by gating out deteriorated cells (conform figure 2 A) on basis of PI staining. It would be good to include the pattern of deteriorated (PI+) cells. For instance do such PI+ cells have a S4 pattern?

REPLY: We apologize for any confusion, all PI+ and PI- cells are included in analysis with no sorting based on PI status. We now specify this in Table 1 legend.

4. It is known from different species including the pig that sperm capacitation induces to a varying extend sperm aggregation. Obviously, this was also observed in the present study (see in the Suppl Information Figure 1 B which depicts multi-sperm events). Although more than one sperm is in the image it is still possible to make a micrograph of all sperm in such an event and detect Zn²⁺ staining patterns in all individual sperm in such micrograph. This information now missing but it is well possible that in such sperm the capacitation specific Zn²⁺ distribution pattern is more frequently present when compared to the single sperm events.

REPLY: We indeed observed sperm aggregation after capacitation; however, aggregation was greatly reduced because we used PVA in our capacitation media, as is customary and described in Materials & Methods. Furthermore, repeated pipetting prior to FlowSight data acquisition completely separated the capacitated spermatozoa, which dissociated any remaining aggregates in a satisfactory manner. Furthermore, we limited our analysis to single cell flow cytometric events; otherwise, it would not be possible to attribute fluorescence intensities to individual cells within each aggregate, when aggregates were present. Most events gated as part of the ‘multiple sperm’ gate are simply multiple sperm in the same image/event rather than head-to-head aggregates. We have changed the plot to density mapped to be clear events wherein multiple sperm per events are minimal/non-existent.

5. The orientation of sperm is also of interest for the interpretation of sperm Zn²⁺ distribution patterns: The supplementary Figure 1 shows elegantly how out-of-focus correction, lateral positioning alignment and masking steps are done before data analysis. However, as shown in Suppl. Fig 1 d-g and in Figure 1 g (S3 and S1) sperm go through the detecting spot either with their head on top or at the back. This will influence the distribution of light emitted from the passing sperm. It is not clear to me how the authors corrected for this head-first versus head-last orientation of sperm.

REPLY: Based in your query, we confirmed with engineers at AMINS (the manufacturers of FlowSight) that head-tail direction does not change intensity readings and image acquisition.

6. The changes in relative proportions of S1-S4 Zn²⁺ distribution patterns have not been correlated to other sperm capacitation responses in this manuscript. I wonder how these changes correlate with the induction of tyrosine phosphorylation, to hyper-activated sperm motility patterns and to the degree of artificial induction of acrosome reaction (i.e. after a Ca²⁺ ionophore or a lysoPC challenge). All these responses are considered to be signs of sperm capacitation.

REPLY: Thank you for suggestions. We have included acrosomal remodeling and acrosomal exocytosis analysis (Figure 2i and Table 2), Zn signature of hyperactivated spermatozoa (Video 1), and Western blot to show tyrosine phosphorylation (SD Figure 5a).

7. The zinc shield idea postulated in figure 3 D (i.e. events happening to sperm termed as signature reversal Zinc spark) are highly speculative and not based on any experimental evidence.

REPLY: We have reduced emphasis on signature reversal, instead to sperm Zn-signaling with inclusion of citations showing Zn effect on chemo-repelling sperm and Zn inhibition of inner acrosomal membrane proteinases acrosin and MMP2, both implicated in sperm-zona pellucida penetration.

8. Likewise this manuscript carries a style of scientific writing allowing too much premature but conclusive statements and trying to explain too many aspects of sperm capacitation/fertilization on only a limited amount of data.

REPLY: We tried to streamline the revised manuscripts, eliminate vague statements and speculations, deemphasized the data from boar fertility trial, provided specific details where needed, and added references in support of particular statements.

Reviewer #3

The experimental results to support the main conclusions (sperm cells with a specific zinc signature indicates higher fertility) remain very preliminary and speculative without clear molecular mechanisms or enough number of samples or replicates in the experiments. In addition, certainly there are areas the authors need to be more clear in their writing. The manuscript also requires more attention to details. For example, there are mislabel and order in the figure or in the figure legend (see the specific comments below).

REPLY: Thank you for your kind comments. We have updated the manuscript with your recommendations, put more emphasis on the molecular mechanisms (added two new paragraphs to that effect) , and moved the mini fertility trial data to supplementary data (as it is not the emphasis of this paper and will be followed up with a larger trial with n>100 with USDA collaborators in the coming year; however due to the nature of the Zn Signature findings and how it prepares spermatozoa for fertilization, we do not want to delay publication by waiting for such future data.

(1) General writing style: Overall, whole manuscript is written very vaguely. For example, in the Abstract, what is “zinc signature”? what is “new paradigm” and what is “older dogma”? The current abstract does not deliver what was done in the study without reading the whole manuscript.

REPLY: We rewrote this manuscript, including abstract. We also addressed the issue of dogmas and paradigms (pls. see response to reviewer 1).

(2) The authors generalize their observation as “mammal” in this manuscript. However, this needs to be more careful. Although Zn inhibits Hv1, the channel does not exist in rodent sperm, for example. This indicates the zinc binding proteins to make Zn signature

would be quite different among the species. Most of all, it is unclear and not demonstrated in the manuscript whether the patterned Zn signature change in sperm is conserved phenomenon in mammalian sperm or present specifically in porcine sperm.
REPLY: We added a supplemental figure showing bull and human sperm Zn distribution similar to that of boar. Also, we added a paragraph discussing possible mechanism by Hv1 and CatSper regulate sperm capacitation, relevant to different mammalian taxa.

(3) General experimental design: In this study, single time point for sperm incubation for 4h is used to see the changes in their Zn signature. How fast is the transition happening during capacitation? Time-laps investigation could help to elucidate the signature change and sperm capacitation in boar.

REPLY: We added supplemental video that shows that signature 2 is associated with hyperactivated sperm motility and signature 3 has reduced motility; the video was acquired within first 30 minutes of in vitro capacitation, showing that changes in Zn signature are rapid.

(4) Line 40-44: This sentence is simply not right. Ca²⁺ entry via CatSper is not a prerequisite for P-Tyr. In fact, it is quite the opposite (Chung et al, Cell (2014), Navarrete et al, J Cell Physiol (2015), Navarrete et al, Sci Rep (2016)).

c We agree that the Ca²⁺ ion role in capacitation differs between rodents and ungulates. In respecting your advice and keeping with our focus on boar as a model animal, we rewrote this sentence as follows: This channel regulates intracellular pH and consequently is thought to be responsible for Ca²⁺ entry via opening of the CatSper channel, all of these events coinciding with PKA activation and the hallmark increase of protein tyrosine phosphorylation during boar sperm capacitation.

(5) Line 46: what are the “certain aspects”? Please be specific.

REPLY: Thank you, this has been elaborated on, with species identification and references added.

(6) Related to Figure 1. Although sperm cells incubated under the capacitating conditions changed the distribution of the distinct patterns of their Zn²⁺ imaging of sperm, there is no direct evidence that the increased fraction (Signature 3) during capacitation is positively correlated with (or causal) to sperm fertility. In particular, sperm cells classified to have signature 3 and 4 group are PI positive. Therefore, it is not clear whether the signature alteration directly reflects sperm capacitation status or is simply due to sperm death and or damage to membrane. To clarify this, it is suggested to directly compare acrosome status and hyperactivated motility with the patterns of Zn distribution after capacitation using different approach other than IBFC.

REPLY: Thank you for your suggestions. We have included Figure 2i and Table 2 to show correlation between Zn Signature and acrosome status as well as Video 1 to show zinc signature of hyperactivated spermatozoa.

(7) Line 72-74: temperature-induced capacitation is over-interpretation. It could be spontaneous. You need room temperature incubation, non-IVC condition required to claim it. Related to Figure 2 and 3. The temperature change in the experiment is over 10

C° (Room temperature, 20 - 25 C° vs capacitation condition, 37 C°). However, the temperature difference between porcine epididymis (36 - 37 C°) and oviduct (38 - 39 C°) is only 2-3 C°. Despite the larger temperature difference than that in vivo status, only around 10 % fraction was shifted from signature 1 to signature 2. And also, the incubation temperature for capacitation (37 C°) is already similar to that in epididymis (36 - 37 C°) where the capacitation is prevented. Therefore, the result should be interpreted again with considering environment during natural mating.

REPLY: Thank you for your comments. Yes, non-IVC samples of the ejaculated spermatozoa were kept at room temperature as mentioned on line 92. Such a protocol is used to emulate the conditions of artificial insemination (>98% swine industry now use AI wherein semen is incubated at reduced temperature prior to mating), as opposed to now seldom used natural mating. We added such wording on this line, as well as sentences discussing the significance of thermal gradients for sperm capacitation and gamete recognition.

(8) Line 77-78: Related to Figure 2 and Extended figure 3. This reviewer believe that the experimental data do not support that majority of ZFN signal is gone signature 3 remain (Figure 2G). Why cell permeable TPEN cannot chelate mitochondrial Zn²⁺?

REPLY: We appreciate the concern. We increased TPEN incubation time to 1 hour and saw an increased amount of Zn probe removed; however, a small population of spermatozoa remained in signature 3 even in the presence of high TPEN. Possible explanation may be that the crosslinking of membranes forming mitochondrial sheath capsule prevents TPEN from actually penetrating in the sperm mitochondria. Accordingly, our serial extraction data (SD Figure SD3j-l) show that these Zn ions can only be removed by detergents at step 3, as oppose to a mere increase of ionic strength at steps 1 & 2. We included a citation showing presence of zinc in the mitochondrial sheath region of dead spermatozoa using electron microscopy.

The authors performed sequential extraction of sperm to investigate how Zn signature is related to membrane structures. However, PBS washing itself induces severe loss of Zn from sperm, followed by 0.75 mM KCl. Is KCl concentration a typo? It is not clear how 0.75 mM KCl after 150 mM NaCl in PBS can tell the ionic effect and the following non-ionic detergent effect on the Zn signature in sperm. Also, in this part, the authors already suggested that simple diffusion could result in depletion of Zn in sperm by PBS washing. Then, why this simple diffusion does not occur to both capacitation and non-capacitation conditions. Other possibility such as cell viability of sperm after PBS washing be excluded in the experiment?

REPLY: We could speculate that Zn-ions are bound to sperm surface by electrostatic force, and thus quickly dissipate from it even with initial PBS wash. Also, the extractions are done in tubes on ice, with such a thermal shock also potentially contributing to Zn-ion efflux.

(9) Lines 465-467: 0. 75 mM KCl treatment?

REPLY: Typo fixed to 0.75 M

(10) Regarding Figure 3. While understanding the difficulty of doing experiments with

big animal models, n=4 is simply not enough to suggest that Zn signature could be used for fertility diagnosis. There is even no difference in Boar B of High Fertility and Boar C of Low fertility in overall Zn signature before IVC. Also there must be a mislabeling in Figure 3b. Boar B should have a higher fold change but the number does not reflect it. Also it will be more straightforward to express b in fold change as Figure 3a is already expressed in percentage. To rectify this point, authors need to increase the number of boars, or do another experiment to investigate fertility, such as in vitro fertilization. Also, correlative analyses could help to support author's suggestion, at least statistic level.

REPLY: We agree that this sample size is too small, for which reason we relegated these preliminary data to SD file. We are following up on this with a larger (n>100 boars) study with collaborators at USDA-ARS Beltsville, MD; however this is not the focal point of this paper and such trial will require much more time. The Pearson's correlation coefficient between fold increase of IVC-induced signature 3 and fertility parameters is 0.97; however, with such a small sample size we do not want to make bold claims until the follow up studies (to be reported separately) are conducted.

There is also no rationale provided why the authors start to combine signature 1 and 2 from Figure 3.

REPLY: This is due to differences to capacitation stimulus responses between pre-rich vs rich sperm fractions. Sperm rich fraction is the one provided by boar studs to breeder for artificial insemination, and to us for boar signature comparisons.

(11) Lines 114-115: This reviewer was not able to find the result corresponding to "Hv1 mRNA abundance has been strongly correlated with male infertility." from the reference the authors cited, not even from the supplementary data.

REPLY: We have updated reference to one showing varying amounts of Hv1 protein in human sperm instead.

(12) Lines 277-278: Again, 0.75 mM KCl used or 0.75M (750 mM) KCl used? In Lines 465-467, KCl concentration was 0.75 mM.

REPLY: Typo corrected to 0.75 M KCl.

Minor Comments:

(1) Line 65: Signature 4, death sperm dead sperm

REPLY: Corrected.

(2) Lines 169-172: the sentence is incomplete – verb missing. "The study of zinc signature many further enhance our understanding of...."

REPLY: Corrected.

(3) Related to Figure 2. Figure legends does not match to figure. Panel numbering should be fixed.

REPLY: Corrected.

(4) Related to Extended figure 2 and 3. In manuscript, extended figure 3 proceeds extended figure 2. The order should be changed. And also, numbers of panels in legends for extended figure 3 should be fixed.

REPLY: Corrected.

Reviewer #4

Overall, study is well-done, but additional data are needed to justify the conclusions.

REPLY: Thank you, we added extensive new data, as detailed above and below.

1. It is known that sperm cells do not capacitate uniformly, and the capacitated sperm population is usually heterogeneous and comprised between 10% to 60% of capacitated sperm cells per fraction. How do authors know that they look at the capacitated spermatozoa? Additional experiments are required to either co-stain spermatozoa with the marker of tyrosine phosphorylation and look at the zinc signature, assess their acrosome status or cherry pick spermatozoa that exhibit hyperactivation as was shown in Chung JJ, Shim SH, Everley RA, Gygi SP, Zhuang X, Clapham DE. Cell. 2014

REPLY: We have included a video file to show the Zinc signature of hyperactivated spermatozoa in Video 1 and included Figure 2i and Table 2 to show the relationship between the zinc signature and capacitation-induced acrosome changes.

2. How do authors exclude the fact that TPEN can also bind calcium (it is not strictly zinc-specific), and one of the reason they see strong staining in the acrosome, is because it is abundant in calcium?

REPLY: We have provided Fluo-4 calcium probe assay, TPEN vs vehicle control, to show 10 uM TPEN for 10 minutes does not chelate Ca^{2+} (SD Figure 4h). Further, we excluded the possibility of FZ3 Zn-probe detecting Ca^{2+} in our experiment since sperm Ca^{2+} levels increases after capacitation while we have shown the FZ3 probe fluorescence decreased concomitantly.

Reviewers' comments:

Reviewer #2 (Remarks to the Author):

This manuscript has been amended after considering the comments made by 4 reviewers. The manuscript has improved significantly.

I still have concern about whether or not the authors did induce full sperm capacitation with the 5 mM bicarbonate levels and omission of phosphodiesterase inhibitors. In our hands such stimulation to boar sperm gave poor IVF results. Either higher bicarbonate levels (15-20 mM) or inclusion of 1 mM caffeine gave good results and may be compared to the present data set.

Reviewer #3 (Remarks to the Author):

Comments to Authors

In general, the authors addressed most of the major and minor criticisms of all reviewers with appropriate experiments and additional data. Although they did not assess directly the transition of Zn distribution during capacitation in time-lapse investigation as suggested in this reviewer's one of major criticisms, they did live imaging and showed a still cut fluorescence together with bright-field video. They commented that this is due to the fast photobleaching of the Zn indicator and long process of capacitation. Importantly, part of the new data (showing motility correlation with signature 1 and 2 and different acrosome status among different Zn signatures) support their hypothesis. Presentation of similar Zn distribution patterns in human spermatozoa also extends the biological significance of their finding.

Minor comments

1. The resolution of figures should be improved than current ones. Especially, labeling in Figure 3a is hard to be recognized.
2. Figure 2g (which shows Zn²⁺ signature under the TPEN treatment) is changed from that in the initial submission. And this picture looks the same to that in Extended data 3h. If it is the duplicate figure, it should be indicated in the figure caption. Also please indicate the meaning of p-value in panel 2e – what are compared to which condition?
3. Changes on Zn²⁺ distribution by sperm capacitation: Reviewers asked to clarify that capacitation changes Zn²⁺-distribution patterns in porcine sperm. For this, reviewers suggested to investigate a) capacitation-related characteristics such as hyperactivated motility, acrosome remodeling, and tyrosine phosphorylation (pY) depending on Zn²⁺-distribution pattern in sperm. It was also asked to do b) time-laps investigation to explain signature changes during capacitation.

For a), authors show that acrosome is remodeled much more in signature 3 and 4 groups rather than 1 and 2 groups. In case of hyperactivated motility, authors show motility pattern of sperm first 30 min induction. Please provide the rationale of their choice of time as the authors used 4h induction in this study for capacitation. In supplemented movies, sperm with signature 2 show motile but it is unclear whether the sperm are hyperactivated. Authors should show motility of porcine sperm cells before and after capacitation. In case of pY analysis, it is also not clear because overall signals are quite similar before and after capacitation except one band between 19 and 26 kDa marker.
4. Temperature induced capacitation: As authors mentioned, current animal industry applies low temperature (18-20 C) to store ejaculated semen for AI. However, as commented in the first

review, physiological temperature can't go down that much, and temperature changes very mildly from male reproductive tract to fertilization site. Therefore, it is not suitable to argue that capacitation is induced by temperature especially in the context of thermotaxis of sperm. Rather, discussion should focus on the application of their finding in industry.

5. Lines 25-26: 'a never before described biological phenomenon' can be simply 'a novel biological phenomenon'.

6. Lines 28: 'zinc signature' term is used before it is defined in line 30.

7. Lines 33: ", inhibited under proteasome inhibiting IVC conditions," Does the first 'inhibited' mean 'reduced'?

8. Lines 37-38: 'lends support'?

9. Lines 38: is the word 'paradigm' appropriate? It could simply state 'support a new role of zinc ions...'

10. Line 56: spell-out UPS when first introduced, not in much later of Line 122.

11. Line 164: reference?

12. Line 165: Does the authors mean 'localized at'

13. Line 174-176: The cited article observed effect of Zn²⁺, Arg, and Coenzyme Q10 to sperm when the compounds are treated together. The article can't explain the effect to sperm when Zn²⁺ is treated solely. Therefore, this citation is not appropriate.

14. Lines 200-204: Break the sentence into two for clarity.

15. Line 224: it should be capital. 'It'

16. Line 233: 'facility'? facilitate?

17. Line 293-294: How authors performed western blot? Authors should give the clear description on how they do the experiment in material and method section.

18. All videos should indicate fps (DIC videos) and the exposure time for the corresponding fluorescent images. Video 1: Please add an arrow to indicate the sperm of interest with Zn signature 2.

Major remaining concerns in response to reviewer #1

1. Possibility of dead sperm in Zn Signature 3 and 4: As reviewer 1 also indicated, the authors used PI to investigate changed membrane status as well as sperm viability in each sperm subpopulation classified by Zn²⁺ distribution. In revised manuscript, authors argue that changed membrane status would cause PI⁺ in Zn-Signature 3 and 4 groups. In the study which is cited by the authors in here, dead sperm showed Zn²⁺ binding to midpiece membrane. In addition, supplemented movie also supports the possibility because Signature 3 spermatozoa do not beat. They might be dead sperm because their mild movement could be due to beating of other sperm tails (Movie 2). Therefore, without knowing proportion of dead sperm in signature 3/4, and viability of individual sperm with differential patterns of Zn²⁺ distribution, current argue can't support authors main hypothesis clearly.

2. Missing data in Figure 2: Relevant to this major concern, the authors answered to reviewer 1 that only PI+ cells show acrosome remodeling and exocytosis in signature 4 group by capacitation, and this result is I on figure 2j. But there is no panel j in figure 2. Did the authors intend to indicate Figure 3C which has 'j' inside the figure.

Reviewer #4 (Remarks to the Author):

the authors addressed my concerns, no additional comments.

RESPONSE TO REVIEW, MS: Zinc Ion Flux during Mammalian Sperm Capacitation

We thank all reviewers for their thoughtful comments which helped us to further strengthen the revised manuscript, which we address in detail below.

Reviewer #2

I still have concern about whether or not the authors did induce full sperm capacitation with the 5 mM bicarbonate levels and omission of phosphodiesterase inhibitors. In our hands such stimulation to boar sperm gave poor IVF results. Either higher bicarbonate levels (15-20 mM) or inclusion of 1 mM caffeine gave good results and may be compared to the present data set.

REPLY: We have shown in our previous paper (Zimmerman et al., 2011) that sperm capacitated under these conditions are able to undergo acrosomal exocytosis induced by solubilized ZP, a key indicator of capacitation. In our hands, sperm capacitated under these conditions prior to incubation with oocytes in mTBM (media absent of 15 mM sodium bicarbonate or caffeine) support normal fertilization rates and high rates of development to blastocyst (Yi, Y.J. et al., 2012 Intl J Androl). In previous revision of the present paper, we showed that such experimental IVC protocol reliably induces hallmark tyrosine phosphorylation (by Western blot), equal to IVC with 15 mM bicarbonate. To be absolutely sure that we have satisfied your concern, we re-ran further comparisons now shown in Extended Data Figure 5c-f. These results uniformly show the same IVC-induced endpoint changes. The main difference is that 15 mM sodium bicarbonate IVC medium capacitates sperm and leads to cell death somewhat faster, a common criticism of such *in vitro* protocol when compared to what occurs *in vivo*. Thus, our IVC media was designed to mimic the sustained, sequential capacitation that is seen *in vivo* compared to IVC medium that forces capacitation quickly.

Reviewer #3

In general, the authors addressed most of the major and minor criticisms of all reviewers with appropriate experiments and additional data. Although they did not assess directly the transition of Zn distribution during capacitation in time-lapse investigation as suggested in this reviewer's one of major criticisms, they did live imaging and showed a still cut fluorescence together with bright-field video. They commented that this is due to the fast photobleaching of the Zn indicator and long process of capacitation. Importantly, part of the new data (showing motility correlation with signature1 and 2 and different acrosome status among different Zn signatures) support their hypothesis. Presentation of similar Zn distribution patterns in human spermatozoa also extends the biological significance of their finding.

REPLY: Thank you for much appreciated positive comments and for understating the constraints of our model system.

Minor comments

3. Changes on Zn²⁺ distribution by sperm capacitation: Reviewers asked to clarify that capacitation changes Zn²⁺-distribution patterns in porcine sperm. For this, reviewers suggested to investigate a) capacitation-related characteristics such as hyperactivated motility, acrosome remodeling, and tyrosine phosphorylation (pY) depending on Zn²⁺-distribution pattern in sperm. It was also asked to do b) time-lapse investigation to explain signature changes during capacitation.

REPLY: As this reviewer already noted above, all those points have been addressed to our best capability in previous revision, with the exception of time-lapse study which is now included in the revised manuscript.

For a), authors show that acrosome is remodeled much more in signature 3 and 4 groups rather than 1 and 2 groups. In case of hyperactivated motility, authors show motility pattern of sperm first 30 min induction. Please provide the rationale of their choice of time as the authors used 4h induction in this study for capacitation.

REPLY: The rationale for this was based on our pilot studies. We initially observed a major Zn signature shift at 4 hours and by time lapse Zn-cytometry, we eventually identified the 30 min time point as the point of transition between sig. 1 and 2. We added this rationale in the manuscript, page 6 and have included time-lapse Zn-cytometry data in Extended Data Fig 3m.

In supplemented movies, sperm with signature 2 show motile but it is unclear whether the sperm are hyperactivated. Authors should show motility of porcine sperm cells before and after capacitation. In case of pY analysis, it is also not clear because overall signals are quite similar before and after capacitation except one band between 19 and 26 kDa marker.

REPLY: We have included an additional supplementary video 1 to show differences between non-IVC, experimental IVC, and 15 mM sodium bicarbonate containing IVC medium porcine spermatozoa motility/hyperactivation. Yes, reviewer is correct: unlike murine or rodent tyrosine phosphorylation, porcine tyrosine phosphorylation is much more modest, without as much prominent changes during the course of capacitation. Therefore new bands after capacitation occur only at the molar weights of 32 kDa (acrosin binding protein), and 21 kDa protein (Phospholipid hydroperoxide glutathione peroxidase). Our results are in accordance with previously reported ones, see for instance Flesch et al (1999) BBRC 262: 787-792; Tardif et al (2001) Biol Reprod 65(3):784-92; Dube et al (2003) J Androl 24:727-33. We added this explanation to MS methods section.

4. Temperature induced capacitation: As authors mentioned, current animal industry applies low temperature (18-20 C) to store ejaculated semen for AI. However, as commented in the first review, physiological temperature can't go down that much, and

temperature changes very mildly from male reproductive tract to fertilization site. Therefore, it is not suitable to argue that capacitation is induced by temperature especially in the context of thermotaxis of sperm. Rather, discussion should focus on the application of their finding in industry.

REPLY: We changed discussion accordingly.

1. The resolution of figures should be improved than current ones. Especially, labeling in Figure 3a is hard to be recognized.
5. Lines 25-26: 'a never before described biological phenomenon' can be simply 'a novel biological phenomenon'.
6. Lines 28: 'zinc signature' term is used before it is defined in line 30.
7. Lines 33: ", inhibited under proteasome inhibiting IVC conditions," Does the first 'inhibited' mean 'reduced'?
8. Lines 37-38: 'lends support'?
9. Lines 38: is the word 'paradigm' appropriate? It could simply state 'support a new role of zinc ions...'
10. Line 56: spell-out UPS when first introduced, not in much later of Line 122.
11. Line 164: reference?
12. Line 165: Does the authors mean 'localized at'
13. Line 174-176: The cited article observed effect of Zn²⁺, Arg, and Coenzyme Q10 to sperm when the compounds are treated together. The article can't explain the effect to sperm when Zn²⁺ is treated solely. Therefore, this citation is not appropriate.
14. Lines 200-204: Break the sentence into two for clarity.
15. Line 224: it should be capital. 'It'
16. Line 233: 'facility'? facilitate?
17. Line 293-294: How authors performed western blot? Authors should give the clear description on how they do the experiment in material and method section.
18. All videos should indicate fps (DIC videos) and the exposure time for the corresponding fluorescent images. Video 1: Please add an arrow to indicate the sperm of interest with Zn signature 2.

REPLY: The above items have been addressed as requested in the manuscript.

2. Figure 2g (which shows Zn²⁺ signature under the TPEN treatment) is changed from that in the initial submission. And this picture looks the same to that in Extended data 3h. If it is the duplicate figure, it should be indicated in the figure caption. Also please indicate the meaning of p-value is in panel 2e – what are compared to which condition?

REPLY: Yes, we updated revision 1 histogram from initial submission; initial submission histogram was with 24-hour-old sperm and revision 1 was changed to fresh sperm. Figure 2g is correct, however Extended Data Figure 3h was accidentally duplicated and since been corrected. The figure caption now describes the p-value listed.

Major remaining concerns in response to reviewer #1

1. Possibility of dead sperm in Zn Signature 3 and 4: As reviewer 1 also indicated, the authors used PI to investigate changed membrane status as well as sperm viability in each sperm subpopulation classified by Zn²⁺ distribution. In revised manuscript, authors argue that changed membrane status would cause PI⁺ in Zn-Signature 3 and 4 groups. In the study which is cited by the authors in here, dead sperm showed Zn²⁺ binding to midpiece membrane. In addition, supplemented movie also supports the possibility because Signature 3 spermatozoa do not beat. They might be dead sperm because their mild movement could be due to beating of other sperm tails (Movie 2). Therefore, without knowing proportion of dead sperm in signature 3/4, and viability of individual sperm with differential patterns of Zn²⁺ distribution, current argue can't support authors main hypothesis clearly.

REPLY: Thank you for bringing the ambiguous interpretation of PI⁺ live vs dead sperm to our attention. We have now included Extended Data Figure 5 (PI time lapse) to illustrate that as the sperm capacitate, they slowly become PI⁺, with two subgroups within the PI⁺ gating: PI⁺ with membrane change and PI⁺ with cell death. We discuss the details on page 7 of revised manuscript.

2. Missing data in Figure 2: Relevant to this major concern, the authors answered to reviewer 1 that only PI⁺ cells show acrosome remodeling and exocytosis in signature 4 group by capacitation, and this result is 1 on figure 2j. But there is no panel j in figure 2. Did the authors intend to indicate Figure 3C which has 'j' inside the figure.

REPLY: Yes, intended Figure 3c in our previous response. The 'j' has been removed from the manuscript figure. One possible interpretation of this is that PI⁺ sperm are those that have membrane changes occur, allowing PI access to the cell nucleus, and acrosome changes do not occur until after these membrane changes.

Reviewer #4 (Remarks to the Author):

The authors addressed my concerns, no additional comments.

REVIEWERS' COMMENTS:

Reviewer #2 (Remarks to the Author):

The authors have satisfactorily addressed my remaining concerns regarding the potential of the medium used to induce functional sperm capacitation allowing proper potential of sperm to fertilize in vitro.

Reviewer #3 (Remarks to the Author):

The authors addressed all the reviewers' concerns. I have only minor comments to help the manuscript for clarity.

1. Line 42-44 should be revised for clarity. "A zinc spark is induced by is..." sentence has two verb
2. Line 52: ", and" should be removed. "from ... to ...zona pellucida including"
3. Line 53-57: please break the sentence
4. Figure 1, the overlay of Signature 1 sperm picture (bright field and fluorescence) is not well aligned. Are the images from the same cell?
5. Line 167-169: this sentence is grammatically not correct. Please revise. "Boars....had..... able to prevail..." ?
6. Line 217: remove comma (,)

REVIEWERS' COMMENTS:

Reviewer #2 (Remarks to the Author):

The authors have satisfactorily addressed my remaining concerns regarding the potential of the medium used to induce functional sperm capacitation allowing proper potential of sperm to fertilize in vitro.

RESPONSE: Thank you, much appreciated.

Reviewer #3 (Remarks to the Author):

The authors addressed all the reviewers' concerns. I have only minor comments to help the manuscript for clarity.

1. Line 42-44 should be revised for clarity. "A zinc spark is induced by is..." sentence has two verb
2. Line 52: ", and" should be removed. "from ... to ...zona pellucida including"
3. Line 53-57: please break the sentence
5. Line 167-169: this sentence is grammatically not correct. Please revise. "Boars....had..... able to prevail...." ?
6. Line 217: remove comma (,)

RESPONSE: Thank you, much appreciated, corrections have been made accordingly.

4. Figure 1, the overlay of Signature 1 sperm picture (bright field and fluorescence) is not well aligned. Are the images from the same cell?

RESPONSE: Yes, the separate channels are of the same cell. All channels are acquired of live spermatozoa, thus sperm motility will move the cell slightly between acquisition of brightfield and fluorescent channels. Such is more pronounced in the whole sperm sample population. We added the following description to the figure caption: "Imprecise fluorescent to brightfield overlay illustrates motile status."